# A molecular mechanism to diversify Ca²⁺ signaling downstream of Gs protein-coupled receptors

Julian Brands[1,2,15], Sergi Bravo [1,15], Lars Jürgenliemke [1,3,15], Lukas Grätz [4], Hannes Schihada[5], Fabian Frechen [6], Judith Alenfelder [1], Cy Pfeil[1,2,12], Paul Georg Ohse[1], Suzune Hiratsuka[7], Kouki Kawakami [7,13], Luna C. Schmacke[5], Nina Heycke[1], Asuka Inoue [7,8], Gabriele König[9], Alexander Pfeifer[10], Dagmar Wachten [6], Gunnar Schulte [4], Torsten Steinmetzer[5], Val J. Watts[11], Jesús Gomeza [1,16], Katharina Simon[1,14,16] & Evi Kostenis [1,16] ✉

A long-held tenet in inositol-lipid signaling is that cleavage of membrane phosphoinositides by phospholipase Cβ (PLCβ) isozymes to increase cytosolic Ca²⁺ in living cells is exclusive to Gq- and Gi-sensitive G protein-coupled receptors (GPCRs). Here we extend this central tenet and show that Gs-GPCRs also partake in inositol-lipid signaling and thereby increase cytosolic Ca²⁺. By combining CRISPR/Cas9 genome editing to delete Gαs, the adenylyl cyclase isoforms 3 and 6, or the PLCβ1-4 isozymes, with pharmacological and genetic inhibition of Gq and G11, we pin down Gs-derived Gβγ as driver of a PLCβ2/3-mediated cytosolic Ca²⁺ release module. This module does not require but crosstalks with Gαs-dependent cAMP, demands Gαq to release PLCβ3 auto-inhibition, but becomes Gq-independent with mutational disruption of the PLCβ3 autoinhibited state. Our findings uncover the key steps of a previously unappreciated mechanism utilized by mammalian cells to finetune their calcium signaling regulation through Gs-GPCRs.

Calcium ions and cAMP are among the most widely used second messengers in mammalian signal transduction[1–4]. The universality of both messengers is best illustrated by their fundamental contribution to a myriad of biological processes as diverse as hormone and neurotransmitter release, muscle contraction, synaptic transmission, metabolism and bioenergetics, gene transcription, and ultimately cell death[1,2,4]. Activated Gs-GPCRs give rise to both mediators, i.e., rapidly convert ATP into cAMP after Gαs-dependent stimulation of adenylyl cyclases (AC), and/or increase cytosolic Ca²⁺ levels either by release from endoplasmic reticulum (ER) stores or by influx from the

[1]Molecular, Cellular and Pharmacobiology Section, Institute for Pharmaceutical Biology, University of Bonn, Bonn, Germany. [2]Research Training Group 1873, University of Bonn, Bonn, Germany. [3]Research Training Group 2873, University of Bonn, Bonn, Germany. [4]Department of Physiology and Pharmacology, Karolinska Institutet, Stockholm, Sweden. [5]Department of Pharmaceutical Chemistry, Philipps-University Marburg, Marburg, Germany. [6]Institute of Innate Immunity, Medical Faculty, University of Bonn, Bonn, Germany. [7]Graduate School of Pharmaceutical Sciences, Tohoku University, Sendai 980-8578, Japan. [8]Graduate School of Pharmaceutical Sciences, Kyoto University, Kyoto 606-8501, Japan. [9]Institute for Pharmaceutical Biology, University of Bonn, Bonn, Germany. [10]Institute of Pharmacology and Toxicology, University Hospital, University of Bonn, Bonn, Germany. [11]Department of Medicinal Chemistry and Molecular Pharmacology, Purdue Institute of Drug Discovery, Purdue University, West Lafayette, IN, USA. [12]Present address: Amsterdam Institute for Molecular and Life Sciences (AIMMS), Division of Medicinal Chemistry, Faculty of Science, Vrije Universiteit Amsterdam, Amsterdam, Netherlands. [13]Present address: Komaba Institute for Science, The University of Tokyo, Meguro, Tokyo 153-8505, Japan. [14]Present address: Department of Pharmaceutical and Pharmacological Sciences, University of Padova, 35131 Padova, Italy. [15]These authors contributed equally: Julian Brands, Sergi Bravo, Lars Jürgenliemke. [16]These authors jointly supervised this work: Jesús Gomeza, Katharina Simon, Evi Kostenis. ✉e-mail: kostenis@uni-bonn.de

extracellular space[5–10]. Given the manifold biological responses resulting after the synthesis of cAMP or elevation of cytosolic $Ca^{2+}$, the number of cellular mechanisms to specifically regulate their intracellular abundance is expected to be rather diverse.

A puzzling feature of Gs signal transduction is that cAMP is consistently formed after Gs-GPCR activation across cells and receptors, whereas Gs-$Ca^{2+}$ is much more variably observed[11–14]. These observations indicate that cAMP may not always be the driving force for Gs-$Ca^{2+}$ and hint at the existence of additional cAMP-independent $Ca^{2+}$ release mechanisms. However, the majority of known Gs-GPCR $Ca^{2+}$ release pathways involve the α subunit of Gs and are, therefore, strictly cAMP-dependent. Among these is the activation of protein kinase A (PKA), a main cAMP effector that phosphorylates L-type calcium channels in cardiomyocytes[6,10], cAMP-EPAC-dependent activation of phospholipase (PL)Cε[8], which enhances cytosolic calcium in cardiac myocytes through $Ca^{2+}$-induced $Ca^{2+}$ release[15], and cAMP-mediated sensitization of $IP_3$-gated ion channels, which release $Ca^{2+}$ from the ER[16–18].

Gs-dependent, cAMP-independent mobilization of ER-$Ca^{2+}$ has also been observed for the β₂-adrenoceptor (β₂AR) in non-excitable cells, which utilizes a molecular mechanism that relies on transactivation of purinergic Gq-coupled P2Y receptors[9]. In this transactivation paradigm, Gs-GPCR-triggered release of ATP into the extracellular space is conditional to the subsequently observed Gq-mediated $Ca^{2+}$ signal. However, a number of independent studies report Gs-$Ca^{2+}$ to require prior Gq-GPCR activation, and hence the signaling hierarchy appears reversed: Gq activation is conditional to Gs-GPCR $Ca^{2+}$ [7,16,19,20].

Canonically, Gq family proteins activate PLCβ1–4 isozymes to catalyze the hydrolysis of the membrane phospholipid phosphatidylinositol-4,5-bisphosphate ($PIP_2$) into membrane-localized DAG and soluble $IP_3$, the latter mobilizing $Ca^{2+}$ from ER stores[21,22]. While this mechanism explains Gs-GPCR $Ca^{2+}$ via transactivation of purinergic P2Y receptors[9], it fails to explain why Gq-GPCR activation is conditional to Gs-$Ca^{2+}$ in non-excitable cells[7,16,19,20].

The prevailing theory is that hydrolysis of $PIP_2$ by PLCβ isozymes to acutely increase intracellular $Ca^{2+}$ is stimulated by both active $Gα_q$ and Gi-liberated Gβγ dimers, the latter of which activate PLCβ2 and PLCβ3 only, but not by active $Gα_s$, Gs-derived Gβγ or $Gα_i$ proteins[21,23–27]. Therefore, the Gβγ-PLCβ-$Ca^{2+}$ signaling axis is generally considered Gi-specific[28–31]. We have recently shown that in a number of mammalian cells from different origins, Gi-liberated Gβγ is insufficient to mobilize $Ca^{2+}$ from ER stores unless active $Gα_q$ provides the licensing trigger[32]. In other words, Gi-Gβγ-PLCβ-$Ca^{2+}$ signals entirely depend on active Gq in mammalian cells[19,28,32,33]. Would active Gq similarly license PLCβ isozymes to become susceptible to Gs-liberated Gβγ? This conjecture—while plausible at first glance—is at odds with a number of independent experimental observations[21,23,24,34–37], suggesting that Gs proteins do not fulfill the criteria for Gβγ signaling. However, the Gq-dependence of Gs-$Ca^{2+}$ signals clearly suggests that Gs-Gβγ signaling may indeed exist and even be physiologically relevant.

In the present study, we provide evidence for both. We show, using CRISPR/Cas9 genome-editing and pharmacological perturbations, that Gs-GPCRs—via Gs-derived Gβγ—partake in inositol-lipid signaling by providing the key mediator—$Ca^{2+}$—for mammalian signal transduction. We classify this Gs-Gβγ-PLCβ-$Ca^{2+}$ module as functionally distinct from that produced by activated Gi, and as independent of but susceptible towards crosstalk with $Gα_s$-promoted cAMP. Thereby, we uncover a long-overlooked mechanism of how Gs-GPCRs contribute to one of the most widely used signal transmission systems—PLCβ-$Ca^{2+}$—in eukaryotic cells, previously considered to be exclusive to Gq- and Gi-GPCRs.

## Results

### Gs-GPCRs mobilize intracellular calcium only after activation of Gq

Mobilization of intracellular $Ca^{2+}$ in non-excitable cells is a hallmark feature of Gs-coupled GPCRs[7,9,38], but the underlying molecular details are poorly understood. To resolve these molecular details, we used the β₂AR, a well-established class A GPCR prototype, as a model. HEK293 cells endogenously express β₂AR making them a useful system to study signaling effects in the absence of overexpression[39,40]. In agreement with active β₂AR signaling, isoproterenol (Iso), a nonselective β-adrenergic agonist, elicited concentration-dependent cAMP formation (Supplementary Fig. 1) but did not produce detectable $Ca^{2+}$ transients (Fig. 1a_i), indicating that the β₂AR-cAMP response is not sufficient to mobilize $Ca^{2+}$ in this cellular background. These observations are in apparent contrast with elegant earlier studies suggesting a cAMP-PLCε-$Ca^{2+}$ release pathway[8,41] or transactivation of nucleotide P2Y receptors as downstream β₂AR event in non-excitable cells[9,40]. Because neither of the above mechanisms requires priming by heterologous Gq-coupled GPCRs[8,9], also proposed by some as an essential element for Gs-$Ca^{2+}$[7,16,19,20,42], we hypothesized that additional mechanisms must exist to promote Gs-$Ca^{2+}$ in non-excitable cells. Indeed, priming of cells with ATP to activate endogenous Gq-coupled P2Y receptors elicited a robust first calcium spike followed by discernable and concentration-dependent β₂AR-induced calcium signals (Fig. 1a_ii). These Gq-primed, Iso-triggered calcium transients were unaltered by pertussis toxin (PTX)-pretreatment ruling out Gi/o contribution (Supplementary Fig. 2). Gq-primed Iso-$Ca^{2+}$ was undetectable in HEK293 cells lacking $Gα_s$ and $Gα_{olf}$ (hereafter HEK-ΔGs[9]) even after Gq priming, uncovering Gs as essential mediator of the observed $Ca^{2+}$ signals (Fig. 1a_iii) and consistent with the absence of Gi/o contribution. Pretreatment of cells with the Gq/11/14-specific inhibitor FR900359 (FR)[43] eliminated the ATP-stimulated calcium response and, consequently, the Iso-mediated calcium response as well (Fig. 1a_iv). Akin to ATP priming, Carbachol (CCh) priming to activate endogenous Gq-coupled muscarinic M3 receptors also enabled Iso-triggered calcium signals in a Gq- and Gs-dependent manner (Fig. 1b). Similar results were obtained with two other Gs-GPCR stimuli acting via endogenous prostanoid $EP_2/EP_4$ and adenosine $A_{2A}/A_{2B}$ receptors: Gs-GPCRs were functional in relaying agonist stimulation to cAMP production (Supplementary Fig. 3), but Gs activation did not suffice to mobilize $Ca^{2+}$ from intracellular stores unless cells were primed with a Gq stimulus (Fig. 1c, d). These latter data were collected in PTX-pretreated cells to ensure that ligand responses were not Gi/o-mediated. In all instances, FR pretreatment but not $Gα_s$ deletion exclusively blunted the Gq-mediated first calcium peak, while a calcium ionophore produced $Ca^{2+}$ rises in parallel experiments across all cell lines and treatment conditions, attesting an intact non-receptor calcium response (Fig. 1a–d, Supplementary Fig. 4). Taken together, our data suggest that Gs-GPCR calcium but not Gs-cAMP requires both functional Gs and Gq and is entirely reliant on Gq priming.

### Gs-calcium demands Gq input in primary cells

A fundamental feature and ubiquitous phenomenon of cell signaling is context-dependence[44–46]. Therefore, we asked whether and to what extent activated Gq is mandatory for Gs-$Ca^{2+}$ in an endogenous signaling environment. We selected murine brown pre-adipocytes (preACs) and mouse embryonic fibroblasts (MEFs) as primary cell models. PreACs are non-excitable and express all three βAR subtypes, the prostanoid $EP_4$ receptor and the two adenosine $A_{2A}$ and $A_{2B}$ receptors[47,48]. In line with our HEK cell findings, Iso addition to preACs promoted the formation of cAMP (Supplementary Fig. 5) but did not mobilize detectable calcium unless cells were primed with serotonin (5-HT), a stimulus for Gq-linked 5-HT receptors (Fig. 2a_i, ii). Consistent with Gq-dependence of Gs-$Ca^{2+}$, FR pretreatment completely prevented all $Ca^{2+}$ elevations without impact on those elicited by the calcium ionophore (Fig. 2a_iii, iv). Prostaglandin $E_1$ ($PGE_1$) and the adenosine agonist NECA mimicked the effects of Iso in that detectable $Ca^{2+}$ demanded prior Gq priming (Fig. 2b, c). Equivalent results were obtained in MEFs, in which Iso-mediated $Ca^{2+}$ traces were elicited only after priming with ATP (Fig. 2d) or UTP (Supplementary Fig. 6) despite

**Fig. 1 | Gs-GPCR mobilization of intracellular $Ca^{2+}$ fully depends on active Gq.** In all HEK293 lines, calcium signals were recorded following a two-step addition protocol. This is exemplified in **a** for the $\beta_2AR$. At $t = 20\,s$, either solvent ($a_i$) or Gq stimulus ATP 100 µM ($a_{ii}$–$a_{iv}$) was added, followed by a second addition at $t = 140\,s$ of either Iso or Calcium ionophore A23187. $a_{iv}$ Cells were pretreated with 1 µM of the Gq inhibitor FR. **b**–**d** Concentration-effect curves derived from the maximum calcium response of the second addition of **b** Iso on $\beta_2AR$, **c** $PGE_1$ on prostanoid $EP_2$ and $EP_4$, **d** NECA on $A_{2A}$ and $A_{2B}$ receptors, or A23187 (5 µM) after prior addition of solvent (no priming), ATP (100 µM) or CCh (100 µM). To exclude the contribution of endogenous Gi/o-coupled prostanoid and adenosine receptors to Gs-$Ca^{2+}$, cells were pretreated overnight (16 h) with 100 ng/ml of the Gi/o inhibitor pertussis toxin (PTX). Representative traces are means + SEM, averaged data are mean ± SEM of $n$ biologically independent experiments (**b**: CCh and solvent $n = 3$, ATP $n = 7$; **c**: $n = 3$; **d**: $n = 3$), each performed in duplicate. Source data are provided as a Source Data file.

detectable cAMP formation (Supplementary Fig. 5). PTX was included in all treatment conditions to eliminate a potentially confounding contribution of endogenous Gi/o-GPCRs, which also require Gq priming for effective $Ca^{2+}$ mobilization[19,28,32]. From these results we concluded that Gs-$Ca^{2+}$ requires a Gq-prestimulus also in the endogenous signaling environment.

## Direct Gq activation by Gs-GPCRs bypasses the requirement of heterologous Gq priming

Although a number of studies agree on the necessity of Gq priming for Gs-$Ca^{2+}$[7,11,16,18–20], the origin of the Gq stimulus remains unclear. Specifically, it is unknown whether active Gq must be provided from another Gq-GPCR by heterologous Gq priming or may also originate from the

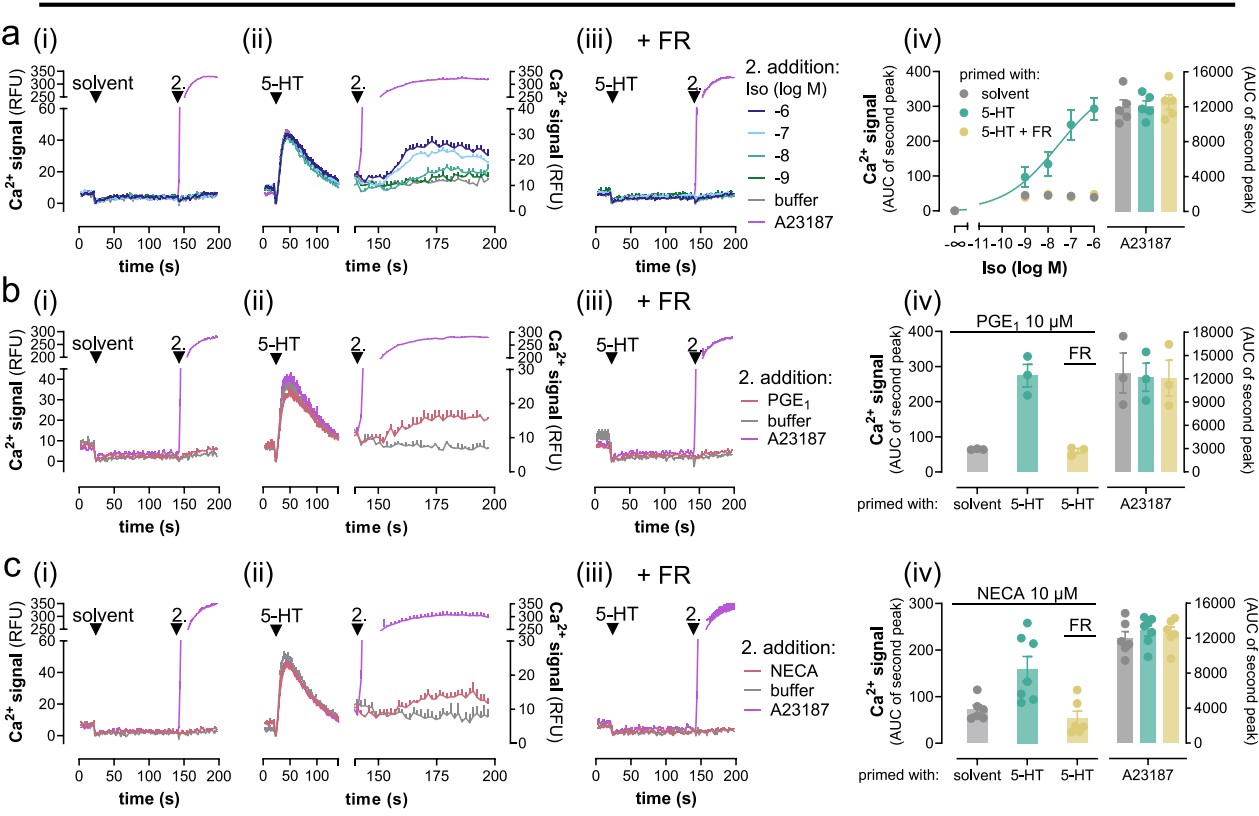

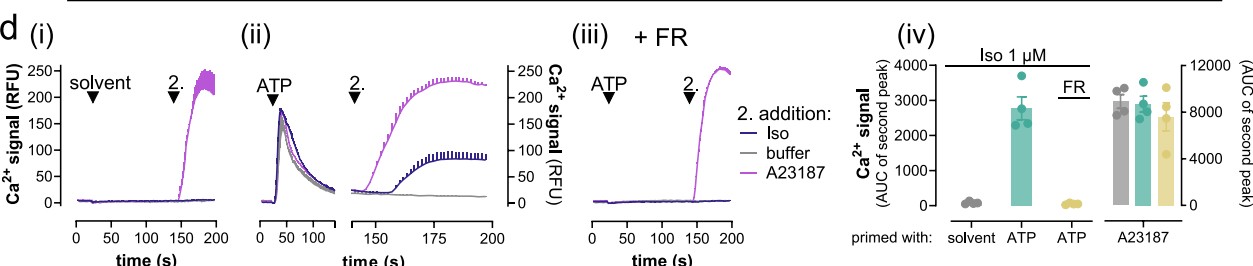

**Fig. 2 | Gs-Calcium demands Gq input in the endogenous signaling context.** Representative calcium recordings and their quantification obtained in primary murine brown pre-adipocytes (preACs) (**a**–**c**) and mouse embryonic fibroblasts (MEFs) (**d**) following a two-step addition protocol. At $t = 20$ s, cells were primed with solvent (**a$_i$**–**d$_i$**), 10 μM 5-hydroxytryptamine (5-HT) (**a$_{ii, iii}$**–**c$_{ii, iii}$**), or 1 μM ATP (**d$_{ii,iii}$**), followed by a second addition at $t = 140$ s of the Gs-GPCR stimuli Iso (**a**), 10 μM PGE$_1$ (**b**), 10 μM NECA (**c**), or 10 μM Iso (**d**) in the presence and absence of 1 μM FR. **a$_{iv}$** Concentration-effect relationships calculated from the data in (**a$_{i–iii}$**) are plotted as the area under the curve (AUC) elicited by Iso stimulation. **b$_{iv}$**–**d$_{iv}$** Bar chart quantification of exemplary data from **b$_{i–iii}$**–**d$_{i–iii}$** including the viability control A23187 (5 μM). Representative recordings are mean + SEM, averaged data are mean ± SEM of $n$ biologically independent experiments (**a$_{iv}$**: $n = 5$; **b$_{iv}$**: $n = 3$; **c$_{iv}$**: solvent and 5-HT + FR $n = 6$, 5-HT $n = 7$; **d$_{iv}$**: $n = 4$), each performed in duplicate. Cells were pretreated with 100 ng/ml of the Gi/o inhibitor PTX 16 h prior to the calcium measurements (**a**–**d**). Source data are provided as a Source Data file.

same Gs-GPCR by direct Gq engagement. To address this question, we studied β$_2$AR calcium signaling in HEK293 cells with enhanced receptor expression, a well-established strategy to facilitate secondary GPCR couplings[49–51]. Indeed, Iso stimulation of overexpressed β$_2$AR promoted robust and concentration-dependent Ca$^{2+}$ transients without prior Gq-stimulation (Fig. 3a$_i$). Because this Ca$^{2+}$ was abolished by FR (Fig. 3a$_{ii, iii}$), and because Gq priming was no longer necessary, we reasoned that overexpressed β$_2$AR may produce its own Gq signal. If this assumption were correct, Gs should no longer be required to elicit β$_2$AR-Ca$^{2+}$. Indeed, Iso-mediated Ca$^{2+}$ transients were detectable in

HEK-ΔGs cells (Fig. 3b$_i$) yet were boosted by Gα$_s$ re-expression (Fig. 3b$_{ii}$) and were fully sensitive to FR pretreatment (Fig. 3b$_{iii, iv}$). We concluded that overexpressed β$_2$AR produces its own Gq signaling sufficient for a Gq-Ca$^{2+}$ response, which explains why the cellular presence of Gs is both dispensable (cf. Fig. 3b$_i$) and conducive (cf. Fig. 3b$_{iii, iv}$) for detection of β$_2$AR-Ca$^{2+}$ in HEK-ΔGs cells. In line with this notion, Gs-independent β$_2$AR-Ca$^{2+}$ was substantially augmented by overexpressed Gα$_q$ and completely blunted by FR (Fig. 3b$_{v,- viii}$), consistent with productive Gq engagement. β$_2$AR expression was comparable across cell lines and transfection conditions, ruling out that

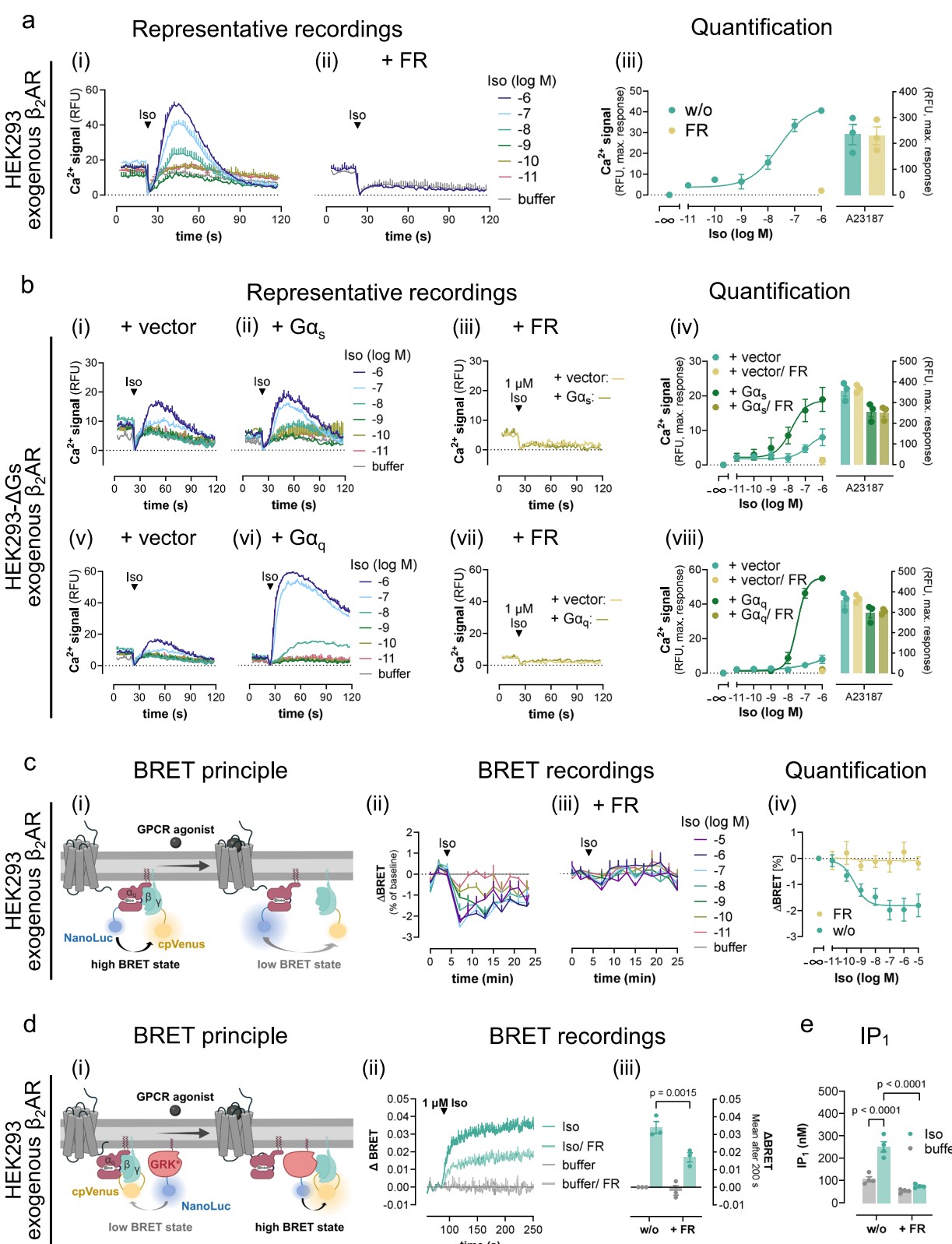

differences in receptor functionality arose from differences in receptor expression (Supplementary Fig. 7). Direct Gq recognition and activation by exogenous $\beta_2$AR is further supported in three distinct ways; with bioluminescence resonance energy transfer (BRET)-based G protein biosensors monitoring activation-induced conformational changes of both modified (Fig. 3c)[52] or unmodified Gq (Fig. 3d)[53], and with IP$_1$ accumulation assays that serve as a proxy for Gq activation

(Fig. 3e). FR completely (Fig. 3c) or partially (Fig. 3d) ablated the detectable BRET changes and fully reversed the Iso-mediated IP$_1$ accumulation (Fig. 3e) confirming direct engagement of Gq by exogenous $\beta_2$AR in all instances. We concluded that $\beta_2$AR mobilizes cytosolic Ca$^{2+}$ in non-excitable cells in a manner, strictly dependent on its cellular abundance: endogenously expressed $\beta_2$AR requires heterologous Gq priming, whereas overexpressed $\beta_2$AR directly engages

**Fig. 3 | Direct Gq coupling of overexpressed $\beta_2$AR eliminates the need for heterologous Gq priming. a, b** Representative $Ca^{2+}$ recordings in response to Iso addition after $t = 20$ s and corresponding quantification of maximum $Ca^{2+}$ peak responses collected in HEK293 (**a**) or HEK-$\Delta$Gs cells (**b**) transfected with the expression plasmid coding for the $\beta_2$AR. HEK-$\Delta$Gs cells were cotransfected with either empty vector DNA (**b$_{i, v}$**), or plasmids coding for G$\alpha_s$ (**b$_{ii, iii}$**), or G$\alpha_q$ proteins (**b$_{vi, vii}$**), respectively. **c$_i$** Cartoon representation of the BRET-based Gq-CASE biosensor which reports separation of G$\alpha_q$ from G$\beta\gamma$ after activation as a decrease of BRET[52]. **c$_{ii-iv}$** Concentration-dependent activation of Gq protein BRET evoked in HEK293 cells with exogenous expression of the $\beta_2$AR and the Gq-CASE biosensor, displayed as real-time BRET recordings and concentration-effect curve derived from the BRET changes after 9 min. **d$_i$** Schematic for the BRET-based G$\beta\gamma$ release assay monitoring freed G$\beta\gamma$ dimers after G protein activation of heterotrimers harboring unmodified G$\alpha$ subunits. **d$_{ii-iii}$** Iso-induced BRET increase between Venus-labeled G$\beta\gamma$ and the membrane-associated C-terminal fragment of the G protein-coupled receptor kinase 3 fused to NanoLuciferase (masGRK3ct-NanoLuc), shown as real-time BRET recordings and their bar chart quantification. **e** Inositol monophosphate (IP$_1$) accumulation measured in naive HEK293 cells transfected to express the $\beta_2$AR. Where indicated, cells were pretreated with FR to silence the function of Gq proteins (1 $\mu$M in **a–d**; 10 $\mu$M in **e**). Representative $Ca^{2+}$ traces and real-time BRET recordings are mean + SEM, averaged data are mean ± SEM of $n$ biologically independent experiments (**a$_{iii}$**: $n = 3$; **b$_{iv}$**: $n = 3$; **b$_{viii}$**: $n = 3$; **c$_{iv}$**: w/o $n = 4$, FR $n = 5$; **d$_{iii}$**: $n = 3$), each performed in duplicate. IP$_1$ accumulation data (**e**) are mean ± SEM of four independent experiments performed in technical triplicates. Statistical significance was calculated with a two-way ANOVA with Fisher´s post-hoc analysis. Source data are provided as a Source Data file. **c$_i$** and **d$_i$**, created with BioRender.com released under a Creative Commons Attribution-NonCommercial-NoDerivs 4.0 International license https://creativecommons.org/licenses/by-nc-nd/4.0/deed.en".

---

Gq, and so bypasses the need of heterologous Gq priming in the HEK293 cellular background.

## Two separable molecular mechanisms account for Gs-GPCR $Ca^{2+}$ after Gq priming

The mechanism of how active Gq coordinates Gs-GPCR calcium is unclear at present. Gs heterotrimers give rise to two separable transducers: G$\alpha_s$, which conveys its signal in a GTP-dependent manner, and G$\beta\gamma$, which activates its effectors by regulated protein-protein interaction[23,32,37,54–57]. Whether G$\alpha_s$-GTP or Gs-derived G$\beta\gamma$ or both are involved in conveying the $Ca^{2+}$ responses after heterologous Gq priming at endogenous $\beta_2$AR expression has not been explored.

G$\alpha_s$-GTP typically stimulates AC isoforms to produce cAMP, which in turn activates two key effectors: PKA and exchange protein directly activated by cAMP (EPAC), both known to increase cytosolic $Ca^{2+}$ by different mechanisms[3,6,8,10,15,41,58–60]. However, neither inhibition of PKA (Fig. 4a) nor of EPAC (Fig. 4b) diminished the Iso-mediated calcium peak amplitudes. These data suggest no major contribution of the two cAMP target proteins to the observed $\beta_2$AR calcium and/or involvement of additional cAMP effectors. cAMP by itself also potentiates Gq-GPCR $Ca^{2+}$ by direct sensitization of inositol-1,4,5-trisphosphate receptors (IP$_3$Rs)[3,16–18]. IP$_3$Rs deliver $Ca^{2+}$ from the ER to the cytosol and other organelles after IP$_3$ binding[1,61–63]. Because IP$_3$ is formed in the priming phase (Supplementary Fig. 8), we probed a direct cAMP-IP$_3$R connection using the AC activator forskolin (Fsk) as a tool to produce cAMP in a GPCR-independent manner. Fsk measurably elevated cAMP within the $Ca^{2+}$ detection window (Supplementary Fig. 9) and mimicked the Iso-mediated $Ca^{2+}$ response after Gq priming regardless of whether ATP or CCh were used as Gq prestimuli (Fig. 4c, d). Thus, cAMP sensitization of IP$_3$Rs may contribute to both Iso and Fsk-induced $Ca^{2+}$ after Gq priming.

Signaling junctions composed of IP$_3$Rs and type 6 AC are responsible for delivering the high cAMP concentrations directly to IP$_3$Rs[3,16–18]. To lower the impact of these junctions and, additionally, the levels of cAMP in response to Gs-GPCR and AC activation, we used HEK293 cells depleted by CRISPR/Cas9 of endogenous AC3 and AC6 (hereafter $\Delta$AC3/6 cells)[64]. Both AC isoforms are highly abundant in HEK293 cells and largely responsible for the Fsk-stimulated cAMP formation in this cellular background[64]. Consistent with a contribution of cAMP to Gs-GPCR and Fsk-$Ca^{2+}$, we observed lower maximal amplitudes and slower kinetics for Iso-$Ca^{2+}$ after Gq priming but no detectable Fsk-$Ca^{2+}$ whatsoever in $\Delta$AC3/6 cells (Fig. 4e, f). These data indicate an altered Iso signaling pattern and, potentially, a molecularly separable, cAMP-independent $Ca^{2+}$ release pathway only for Iso. Interestingly, only the low-potency $Ca^{2+}$ signals (10 nM–1 $\mu$M Iso) were maintained in the $\Delta$AC3/6 cells, while the high-potency $Ca^{2+}$ signals (0.1 nM–10 nM Iso) were essentially eliminated, consistent with their dependence on cAMP and the low cAMP production we observed at maximal $\beta_2$AR or AC stimulation (Supplementary Fig. 10). We concluded that Iso and Fsk share a cAMP-driven component within the complex $Ca^{2+}$ concentration-effect curve, while Iso stands out unique with an additional cAMP-independent yet quantitatively minor $Ca^{2+}$ release mechanism.

## Fsk serves as a proxy to discriminate cAMP-dependent from cAMP-independent $Ca^{2+}$ after Gq priming

To investigate whether a complementary approach to diminish the overall cAMP-IP$_3$R impact would also allow the unmasking contribution of the cAMP-independent $Ca^{2+}$ release mechanism, we employed Gq priming at low stimulus intensity. Indeed, a two-component concentration-effect relationship emerged exclusively for Iso after priming with both CCh and ATP at single-digit micromolar concentrations (Fig. 5a, b). We noted that the Iso-mediated high-potency $Ca^{2+}$ release response was closely resembled in magnitude by Fsk at a maximally effective concentration (Fig. 5a, b). Therefore, we used Fsk as a proxy to probe the contribution of cAMP to the $Ca^{2+}$ release mechanisms engaged by Gs-GPCRs in our primary cell models. Interestingly, unlike Iso-$Ca^{2+}$, which readily emerged after Gq priming in both preACs and MEFs, Fsk-$Ca^{2+}$ was undetectable in the preACs, but detectable in MEFs, yet smaller in amplitude as compared with Iso-$Ca^{2+}$ (Fig. 5c, d and Supplementary Fig. 11). Because robust cAMP formation was observable for both stimuli within the $Ca^{2+}$ detection window under primed and non-primed conditions in both preACs and MEFs (Fig. 5e), and because Fsk-cAMP even surpassed that of Iso in amplitude in preACs (Fig. 5e$_{ii}$), we interpreted that the absence of detectable Fsk-$Ca^{2+}$ in preACs indicates no major contribution of the cAMP-dependent mechanism in this cellular background. Conversely, both cAMP-dependent and cAMP-independent $Ca^{2+}$ release mechanisms are operative in MEFs. From these data, we concluded that (i) the qualitative and quantitative contribution of Gs-GPCR-$Ca^{2+}$ release pathways is cellular context-dependent, and (ii) Iso-$\beta_2$AR-Gs-$Ca^{2+}$ in HEK293 cells (Fig. 5a, b) is composed of two separable molecular mechanisms, one reliant on cAMP and involving sensitization of IP$_3$Rs, the other cAMP-independent but otherwise undefined.

## Ligand-activated Gs-GPCRs drive IP$_3$ formation and PIP$_2$ depletion after Gq priming

Because IP$_3$Rs are a point of convergence for distinct upstream signaling pathways (Gs, Gq, Gi/o-$\beta\gamma$), we explored their involvement using the IP$_3$R antagonist 2-APB. Pretreatment of HEK293 cells and of $\Delta$AC3/6 cells with 2-APB eliminated the Iso-mediated $\beta_2$AR-$Ca^{2+}$ after Gq priming but also all Gq-$Ca^{2+}$ evoked by ATP, indicating that IP$_3$-mediated $Ca^{2+}$ release is an essential step for both stimuli (Fig. 6a, Supplementary Fig. 12). Because Iso Gs-$Ca^{2+}$ requires Gq priming but not the mere elevation of cytosolic $Ca^{2+}$ (Supplementary Fig. 13) and because 2-APB nullifies all IP$_3$R-mediated $Ca^{2+}$ but not Gq activation (Fig. 6b),

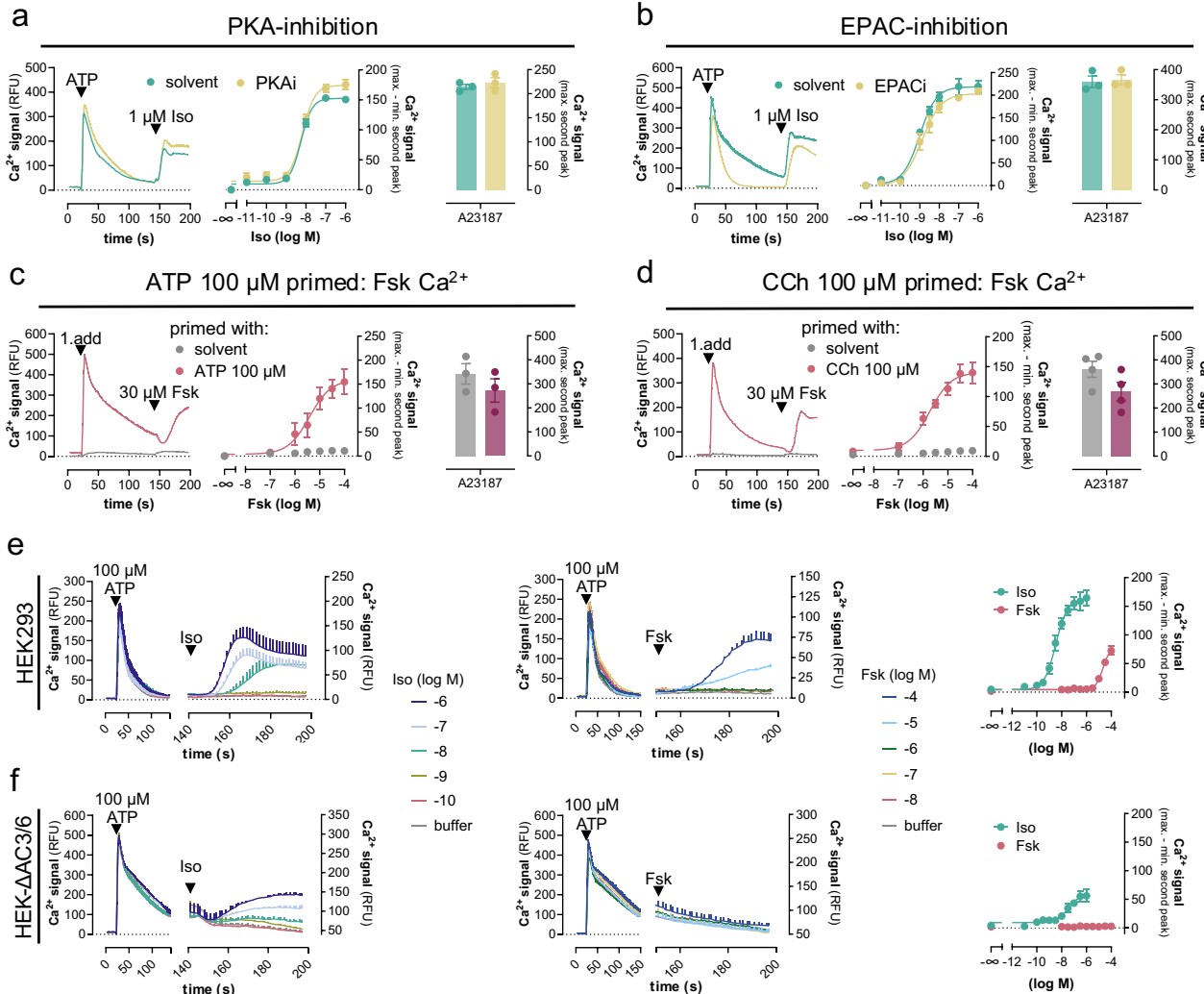

**Fig. 4 | Gs-GPCRs use two separable calcium release mechanisms, both of which depend on prior Gq priming.** Calcium mobilization in HEK293 and HEK-ΔAC3/6 cells following the two consecutive addition protocol. Images show representative real-time Ca²⁺ recordings, concentration-effect curves derived therefrom, and bar chart quantification for the enhancement of cytosolic Ca²⁺ by the calcium iono-phore A23187 (5 μM). **a, b** Ca²⁺ responses in HEK293 cells in the absence or presence of (**a**) 10 μM PKA-inhibitor (PKI14-22) or (**b**) 25 μM EPAC-inhibitor (HJC0197) after

priming with 100 μM ATP and followed by addition of Iso. **c, d** Fsk-Ca²⁺ in HEK293 cells with and without prior Gq priming. **e, f** Iso- and Fsk-induced cytosolic Ca²⁺ increase in HEK293 (**e**) and HEK-ΔAC3/6 cells (**f**) after ATP priming. Representative calcium traces are means + SEM. Quantified data are mean values ± SEM for *n* independent biological experiments (**a–c**: *n* = 3; **d**: *n* = 4; **e**: Iso *n* = 4, Fsk *n* = 9; **f**: *n* = 4), each performed in duplicate. Source data are provided as a Source Data file.

the absence of Gs-Ca²⁺ after 2-APB treatment strongly suggests that the additional cAMP-independent pathway is also IP₃R-dependent.

If this assumption were correct, Gs-GPCRs should ultimately generate the Ca²⁺ mobilizing second messenger IP₃ by themselves. Indeed, monitoring the cellular IP₃ levels in real-time with a con-formational BRET-based IP₃-biosensor[65] revealed Iso-induced IP₃ for-mation exclusively after Gq priming (Fig. 6c). Because IP₃ production is rapid and transient as it is metabolized to IP₂ and IP₁, we also quantified its degradation product IP₁ after accumulation in cells. We detected robust Iso-induced IP₁ accumulation exclusively after Gq priming (Fig. 6d$_i$). We obtained equivalent results for the two other Gs-GPCR stimuli, PGE₁ and NECA, respectively, both provoking IP₁ accumulation only after Gq priming (Fig. 6d$_{ii, iii}$). We also observed an Iso-mediated reduction of PIP₂ levels, the immediate consequence of PLCβ hydro-lysis, in Gq-primed cells, and this effect was completely blunted by FR pretreatment (Fig. 6e). These data point to the active participation of Gs-GPCRs in plasma membrane phospholipid hydrolysis by stimula-tion of PLCβ isozymes, key orchestrators of inositol-lipid-dependent signaling responses.

## PLCβ2 and β3 but not β1 and β4 facilitate Gs-GPCR driven, cAMP-independent Ca²⁺

PIP₂ depletion, IP₃ formation as well as IP₁ accumulation by Gs-GPCRs after Gq priming is a strong indication of signaling activity involving PLCβ isozymes, which hydrolyze PIP₂ into the signaling molecules DAG and IP₃[21,22]. To directly test engagement of this signaling branch by Gs-GPCRs, we engineered HEK293 cells by CRISPR/Cas9 genome editing to nullify functional expression of the PLCβ1–4 isoforms (hereafter HEK-Δ$_f$PLCβ1–4) (Supplementary Fig. 14). This approach allows us to distinguish the regulation of individual isoforms by either Gα$_{q/11}$ and/or Gβγ after re-expression. In agreement with deficient functional expression of PLCβ1–4, Ca²⁺ transients in response to canonical Gq or to dual Gq-Gi stimuli were undetectable in HEK-Δ$_f$PLCβ1–4 cells. Upon re-expression of each individual PLCβ isoform by transient transfec-tion, their expected natural regulation by both Gq/11 (Supplementary Fig. 15) and Gi-liberated Gβγ subunits (Supplementary Fig. 16) was faithfully recovered. These data validate the CRISPR clone for analysis of PLCβ-Ca²⁺ signaling despite the detection of some residual non-KO alleles (cf. Supplementary Fig. 14). Re-expression of each PLCβ isoform

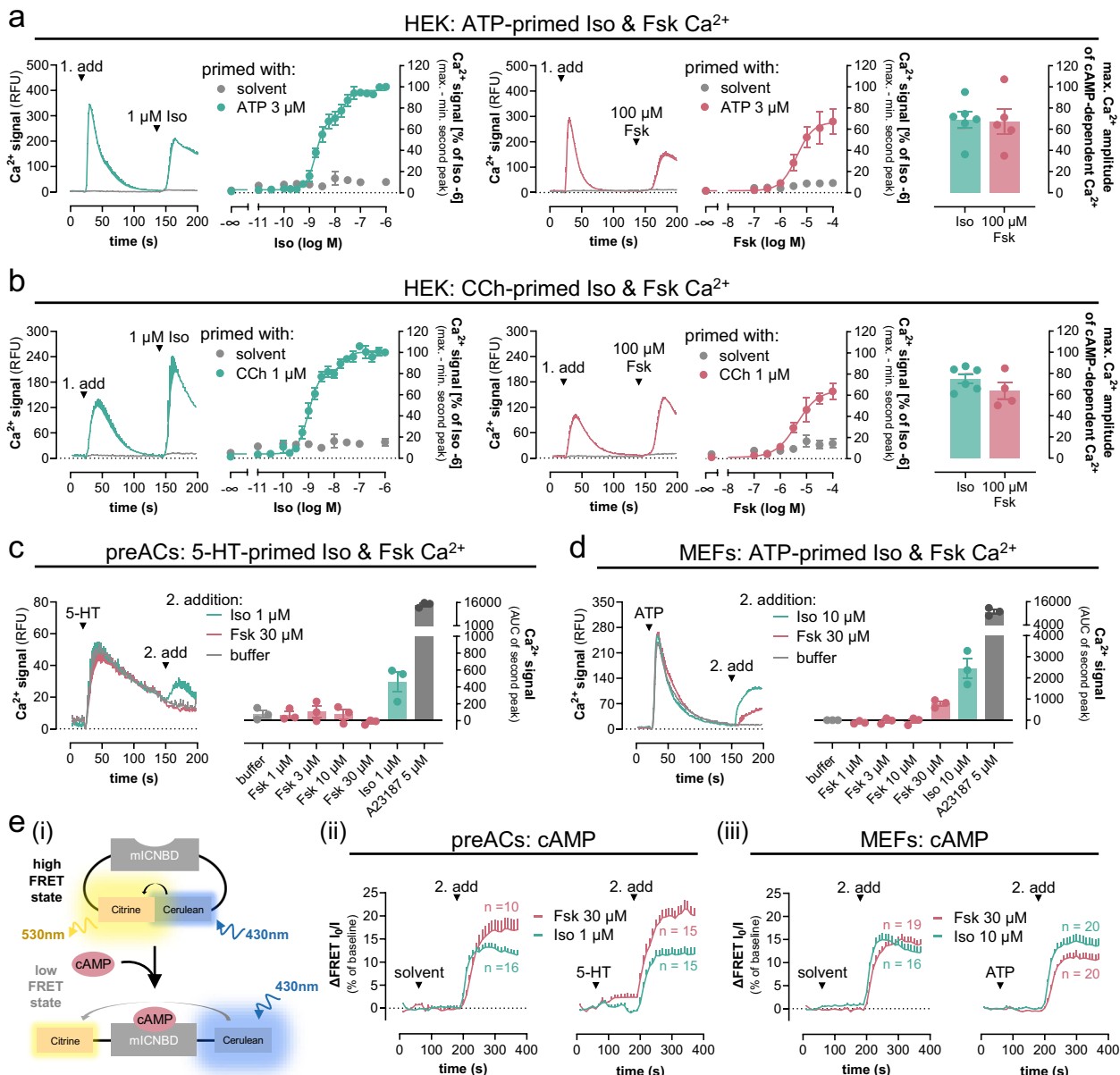

**Fig. 5 | Fsk is a proxy to discriminate cAMP-dependent from cAMP-independent Ca²⁺ after Gq priming in recombinant and primary cells. a–d** Calcium mobilization in HEK293 cells (**a**, **b**), primary pre-adipocytes (preACs, **c**), and MEFs (**d**) following the two consecutive addition protocol. **a**, **b** Iso- and Fsk-induced cytosolic Ca²⁺ increase in HEK293 cells after priming with solvent, 3 μM ATP (**a**) or 1 μM CCh (**b**). **c**, **d** Iso- and Fsk-Ca²⁺ in preACs (**c**) and MEFs (**d**) after priming with 10 μM 5-HT (**c**) or 1 μM ATP (**d**). Data show representative real-time Ca²⁺ recordings and their quantification as either concentration-effect curves (**a**, **b**) or bar charts (**c**, **d**) including the calcium ionophore A23187 (5 μM). The two rightmost panels in **a** and **b** depict the maximum Ca²⁺ amplitudes of the Iso-mediated high-potency Ca²⁺ release response along with Fsk at a maximally active concentration. **e** Live-cell real-time cAMP imaging in preACs (**e_ii**) and MEFs (**e_iii**) using the intramolecular FRET-based pcDNA3.1-mICNDB sensor[103]. **e_i** Cartoon illustration of the sensor principle:

The sensor contains the cyclic nucleotide-binding domain from the bacterial *Mlo-tiK1* channel (mICNBD) flanked by citrine and cerulean at its N- and C-terminus, respectively. At low cAMP abundance both fluorophores are in close proximity (high FRET state) but move further apart upon cAMP increases (low FRET state). FRET changes in response to Iso and Fsk under non-primed and primed conditions in preACs (**e_ii**) and MEFs (**e_iii**) are means + SEM of the indicated n cells. FRET ratios are inverted to show enhanced cAMP abundance as increased FRET ratios. Pooled data are mean values ± SEM of *n* independent biological experiments (**a**: Iso *n* = 6, Fsk *n* = 5; **b**: Iso *n* = 6, Fsk *n* = 4; **c**, **d**: *n* = 3), each performed in duplicate. Representative calcium traces are means + SEM. Data in (**a**, **b**) were fit to a biphasic concentration-effect model to minimize the distance of the measured data points from the predicted data points without using constraints. Source data are provided as a Source Data file.

was also sufficient to re-establish detectable Iso-Ca²⁺ as well as Fsk-Ca²⁺ after Gq priming in an isoform-specific manner. PLCβ1 and β4 enabled detection of a "mono-component high potency" Ca²⁺ release pathway/response, that was mimicked in magnitude by Fsk at a maximally effective concentration (Fig. 7a–c). PLCβ2, on the contrary, enabled mono-component Iso-Ca²⁺ but barely detectable Fsk-Ca²⁺, while PLCβ3 permitted distinction of two pharmacologically separable Ca²⁺ release

responses. The first component (=high-potency response) was observable at single-digit nanomolar Iso concentrations and was similar in magnitude to Fsk; the second component (=low-potency response) occurred at submicromolar Iso concentrations, with a maximal efficacy surpassing that of Fsk (Fig. 7d, e). For any of the PLCβ isoforms, Gs-Ca²⁺ was Gq-dependent (Supplementary Fig. 17). These results provide compelling evidence that PLCβ1 and β4 exclusively support a

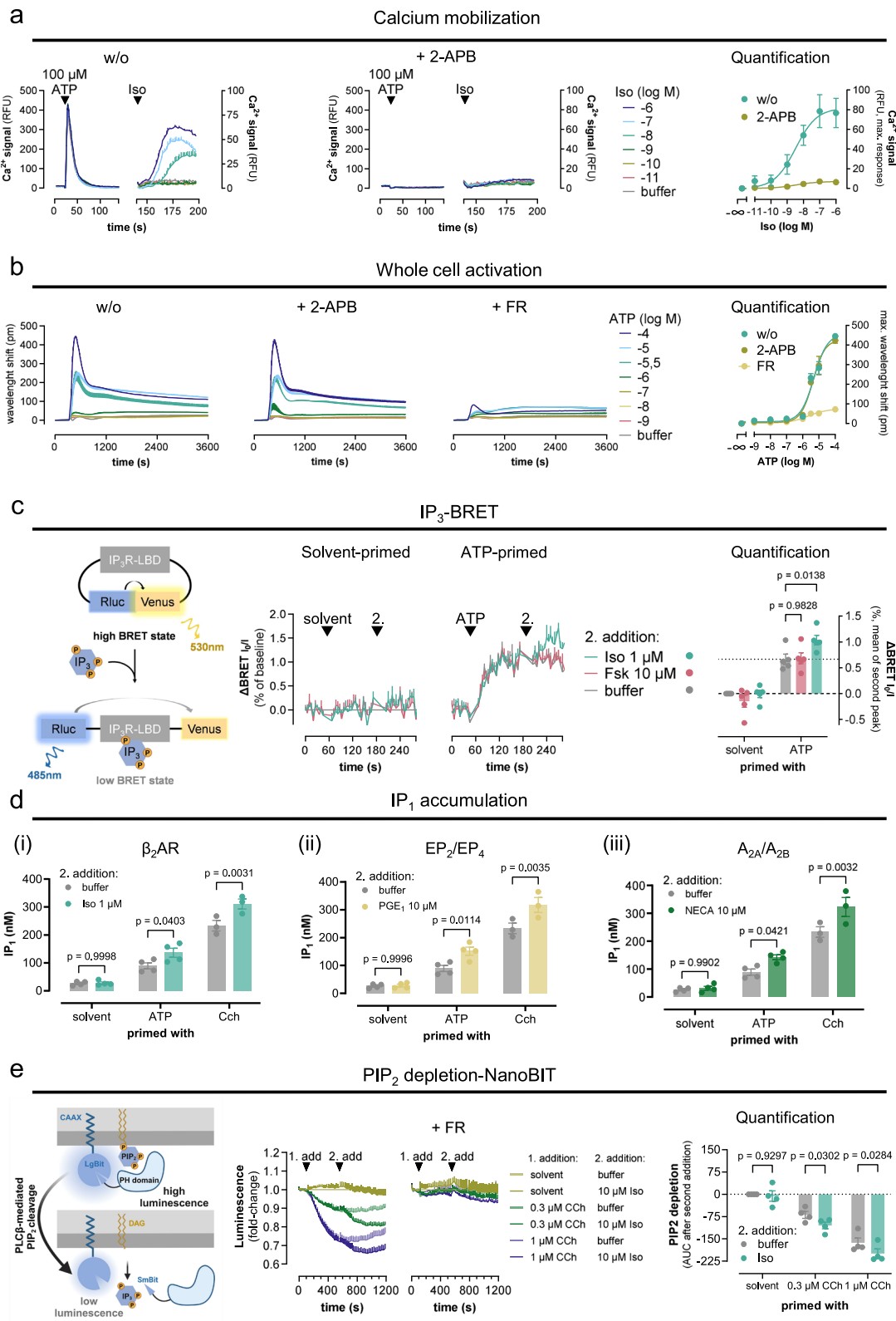

uniform cAMP-driven $Ca^{2+}$ release mechanism, while $PLC\beta2$ enables $Ca^{2+}$ regulation by a Gs-dependent but cAMP-independent pathway only. $PLC\beta3$, on the other hand, empowers dual $Ca^{2+}$ regulation by cAMP and an additional Gs-dependent but cAMP-independent pathway. This qualitative and quantifiable difference between $PLC\beta1$ and $\beta4$ versus $PLC\beta2$ and $\beta3$ isozymes parallels with their natural regulation by G protein $\beta\gamma$ subunits[25,27,33].

## Gs-liberated G$\beta\gamma$ subunits expand the repertoire of direct PLC$\beta$3 activators

To investigate whether Gs-derived G$\beta\gamma$ is responsible for Gs-dependent, cAMP-independent $Ca^{2+}$ after Gq priming, we used the established G$\beta\gamma$ scavenger masGRK3ct, a membrane-associated fragment of the GPCR kinase 3 C-terminus, which does not bind to G$\alpha$ subunits[66–68]. MasGRK3ct visibly diminished the Iso-induced low-potency $Ca^{2+}$

**Fig. 6 | Gs-coupled β₂AR drives IP₃ formation, IP₁ accumulation, and PIP₂ depletion after Gq priming. a** Representative Iso-induced Ca²⁺ traces and their quantification in the absence or presence of 50 μM of the IP₃R antagonist 2-APB in naive HEK293 cells after ATP priming. **b** Exemplary label-free whole cell activation profiles, based on detection of dynamic mass redistribution (DMR) in response to ATP-stimulated Gq-coupled P2Y receptors in untreated (w/o), 2-APB-treated (50 μM), and FR-treated (1 μM) HEK293 cells, and corresponding quantification. **c** BRET-based real-time IP₃ detection following a two consecutive addition protocol. Cartoon illustrating the IP₃ intramolecular BRET biosensor principle[65]. In IP₃-free conditions, energy donor Renilla luciferase (Rluc) and energy acceptor Venus, each fused to the IP₃R ligand binding domain (LBD) are in close proximity (high BRET state). Binding of an IP₃ molecule triggers donor:acceptor separation, resulting in a BRET decrease (low BRET state). BRET ratios are plotted as reciprocals of the $I/I_o$ values. **d$_{i-iii}$** Agonist-induced IP₁ accumulation in HEK293 cells with and without ATP (100 μM) or CCh (100 μM) priming using Iso (**d$_i$**), PGE₁ (**d$_{ii}$**), and NECA (**d$_{iii}$**) to stimulate β₂AR, EP₂/EP₄, and A₂ₐ/A₂ᵦ, respectively. **e** Iso-induced PIP₂ depletion after Gq priming. Schematic of the PIP₂ hydrolysis NanoBiT-based biosensor. PIP₂ hydrolysis is reflected by rapid translocation of the Small BiT (SmBiT)-tagged PH domain of PLCδ1 from plasma membrane-localized Large BiT (LgBiT)-CAAX to the cytosol resulting in decreased luminescence. Real-time recordings in (**a, b**) are mean values + SEM. IP₃ (**c**) and PIP₂ (**e**) recordings, concentration-effect curves (**a, b**), and bar charts (**c-e**) are mean values ± SEM for *n* independent biological experiments (**a**: *n* = 4; **b**: *n* = 3; **c**: *n* = 5; **d**: solvent and ATP *n* = 4, CCh *n* = 3; **e**: *n* = 4). Ca²⁺ measurements are duplicates; DMR, IP₁ accumulation, and PIP₂ depletion are triplicate, and IP₃-BRET time-courses are quadruplicate determinations. Statistical significance was calculated with a two-way ANOVA with Dunnett's (**c**) and Šídák's (**d, e**) post-hoc analysis. Source data are provided as a Source Data file. **c** and **e** was created with BioRender.com released under a Creative Commons Attribution-NonCommercial-NoDerivs 4.0 International license https://creativecommons.org/licenses/by-nc-nd/4.0/deed.en".

release pathway in HEK-Δ_fPLCβ1–4 cells after re-expression of PLCβ3 (Fig. 8a), and in HEK293-wt cells after Gq priming at low stimulus intensity (Fig. 8b). MasGRK3ct also affected CCh-Ca²⁺ but to a lesser extent than Iso-Ca²⁺, and hardly impacted Ca²⁺ transients induced by calcium ionophore A23187 (Supplementary Fig. S18). These data are consistent with efficient sequestration of Gs-derived Gβγ by masGRK3ct across the two cellular systems, and, hence, with Gs-initiated but Gβγ-driven Ca²⁺ signaling via PLCβ3 after heterologous Gq priming.

## Gβγ-regulated PLCβ3 variants bypass the requirement of Gq priming to trigger Gs-Ca²⁺ in living cells

PLCβ enzymes are strictly autoinhibited, and activation is possible only if this autoinhibition is relieved by either Gq or Gβγ subunits[55,69,70]. Because Gq is determinant for Gs-Gβγ Ca²⁺ and for assisting Gs-GPCRs in promoting the PIP₂ hydrolysis reaction, we reasoned that mutant PLCβ3 variants with crippled autoinhibition should empower stand-alone control of Gs-Gβγ-Ca²⁺ in living cells. We used PLCβ3 variants with mutational deletion of highly conserved autoinhibitory elements, PLCβ3^F715A and PLCβ3^ΔXY [69] to test our hypothesis. We used FR-treated cells to pharmacologically silence any possible endogenous Gq activity. Indeed, both PLCβ3 mutants but not PLCβ3-wt, expressed at comparable protein abundance, warranted Iso-mediated Gs-Ca²⁺ without heterologous Gq priming (Fig. 9a and Supplementary Fig. 19). Consistent with direct activation of the two PLCβ3 mutants by Gs-derived Gβγ, Iso-mediated β₂AR-Ca²⁺ was undetectable in Gs null cells (HEK-ΔGs) despite successful expression of all PLCβ3 variants (Fig. 9b and Supplementary Fig. 19). Moreover, Iso-induced Gs-Ca²⁺ but not that induced by Ca²⁺ ionophore was effectively reversed (PLCβ3^ΔXY) or fully blunted (PLCβ3^F715A) by Gβγ sequestration with masGRK3ct (Fig. 9c, d). These data strongly argue for a functional connection between Gs-derived Gβγ and PLCβ-calcium, albeit, such integrated Ca²⁺ transients may obviously be confounded by cAMP-dependent IP₃R sensitization[3,16–18], and hence cAMP-dependent Ca²⁺ in living cells. Indeed, the lower efficacy of masGRK3ct to diminish Iso-Ca²⁺ in PLCβ3^ΔXY as compared with PLCβ3^F715A expressing cells parallels with the stronger constitutive activity of the PLCβ3^ΔXY variant[32,55,69,70], and hence a more prevalent contribution of the cAMP-IP₃R axis[16,17].

To eliminate this confounding variable and to unambiguously isolate the direct activation of PLCβ3 by Gs-derived Gβγ, we quantified PIP₂ depletion, the immediate consequence of PIP₂ hydrolysis as well as the formation of IP₃, an immediate product of the PIP₂ hydrolysis reaction upstream of IP₃R-controlled and ER-liberated Ca²⁺ (Fig. 9e–g). Indeed, Gβγ-regulated PLCβ3^F715A drives both Iso-mediated PIP₂ depletion and IP₃ formation without Gq priming, and these effects were nullified by masGRK3ct (Fig. 9e, f). These measurements were performed in HEK cells genome-edited to lack functional alleles for Gα_q, Gα₁₁, as well as Gα₁₂ and Gα₁₃ (HEK ΔGq/11/12/13)[71], to eliminate

any conceivable contribution by endogenous PLCβ and/or PLCε signaling modules downstream of Gq/11 and G12/13 family G proteins, respectively[8,21,22,41]. In conclusion, our mechanistic dissection of Iso-mediated β₂AR-Ca²⁺ after Gq priming uncovers an additional Ca²⁺ release pathway downstream of Gs-GPCRs to augment their signaling versatility in non-excitable cells. This pathway is Gs-dependent yet cAMP-independent but instead driven by Gs-derived Gβγ after Gq priming. Thereby Gs-GPCRs gain control over the hydrolysis of plasma membrane PIP₂ by PLCβ isozymes, previously considered a conserved property of Gq- and Gi/o-GPCRs only.

## Discussion
### The major finding
Phosphoinositides are low-abundance acidic phospholipids in the cytoplasmic leaflet of all eukaryotic cellular membranes. They are substrates of phosphoinositide-hydrolyzing PLCβ and PLCγ enzymes and the source of central mediators for mammalian signal transduction: IP₃ and Ca²⁺[21,22,72]. A long-held tenet in inositol-lipid signaling is that Ca²⁺ mobilization downstream of activated PLCβ isozymes is mediated by heterotrimeric Gq and/or Gi proteins, using either their Gα_q/11 or their Gi-βγ subunit complexes, respectively[21,23,25,27,73]. The major finding of the present study is that Gs proteins also partake in inositol-lipid-dependent signaling using Gs-liberated Gβγ to ultimately enhance intracellular Ca²⁺. This is even more remarkable considering that Gs heterotrimers are generally considered as poor providers of free Gβγ unlike Gi/o proteins, which are well known as efficient Gβγ donors[19,24,25,28–30,34,36]. Freed Gβγ dimers directly interact with their downstream effectors to initiate a number of signaling events that—together with Gα—define the overall GPCR signaling response[24,74,75].

### Does Gs-liberated Gβγ function as a stand-alone signaling entity?
There is logic and reason as to why Gs-GPCRs when compared to Gi-GPCRs would be much less suited to signal via Gβγ, including lower expression of Gs proteins[21,23], or less efficacious and slower Gα_s-Gβγ dissociation compared with Gi/o heterotrimers[24,35,37,76]. Indeed, a consistent feature of Gs protein biosensors is their much less pronounced conformational rearrangement as compared with equivalent sensors for other G protein families[35,77,78]. Early elegant and recent work even suggests that Gs heterotrimers do not fully dissociate during activation[34,76,77,79] and that GTP hydrolysis may even occur within a non-dissociated Gs heterotrimer[34,80–82]. However, none of these arguments securely rules out Gs-Gβγ as an active signaling entity. To provide but one counterargument, Gs heterotrimers also activate G protein-gated inwardly rectifying potassium (GIRK) channels, which are considered as paradigmatic Gβγ targets via Gs-liberated Gβγ, but do so less effectively as compared with Go heterotrimers[34,36]. Regardless, liberated Gβγ dimers are known to function as highly efficient signal

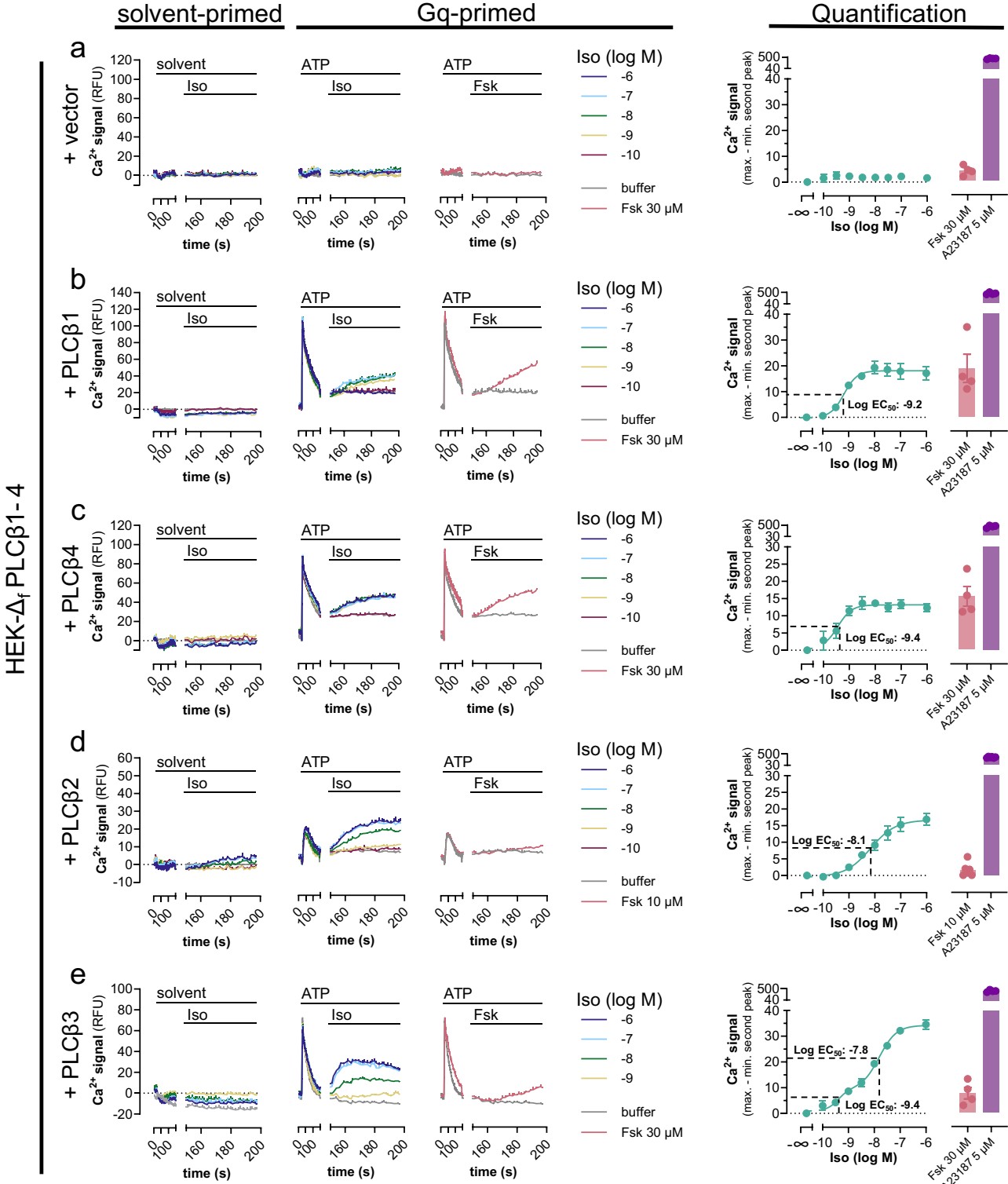

**Fig. 7 | PLCβ2 and β3, but not PLCβ1 and β4, use an additional Gs-dependent, cAMP-independent Ca²⁺ release pathway.** HEK-Δ$_f$PLCβ1–4 cells transiently transfected with either empty expression vector (**a**) or plasmid cDNA coding for each individual PLCβ1–4 isoform (**b**–**e**) were primed with solvent or 100 μM ATP (first addition at $t = 20$ s) followed by a second addition at $t = 140$ s of Iso or Fsk as indicated. Solvent-primed representative Ca²⁺-fluorescence recordings are buffer-corrected, while Gq-primed exemplary Ca²⁺ fluorescence traces are not. Ca²⁺ responses are quantified as concentration-effect curves for net mean peak responses to Iso, or as bar charts for Fsk and calcium ionophore A23187. Inflection points are marked with the corresponding EC₅₀ values. Representative traces are presented as mean values + SEM, averaged data are mean values ± SEM of $n$ biological replicates (**a**–**c**, **e**: $n = 4$; **d**: $n = 6$), each performed in duplicates. Data in **e** were fit to a biphasic concentration-effect model to minimize the distance of the measured data points from the predicted data points. Slope factors nH₁ and nH₂ were constrained to equal 2.0 and 1.3 ($r^2 = 0.97$) respectively. Statistical significance was calculated with a two-way ANOVA with Šídák's post-hoc analysis. Source data are provided as a Source Data file.

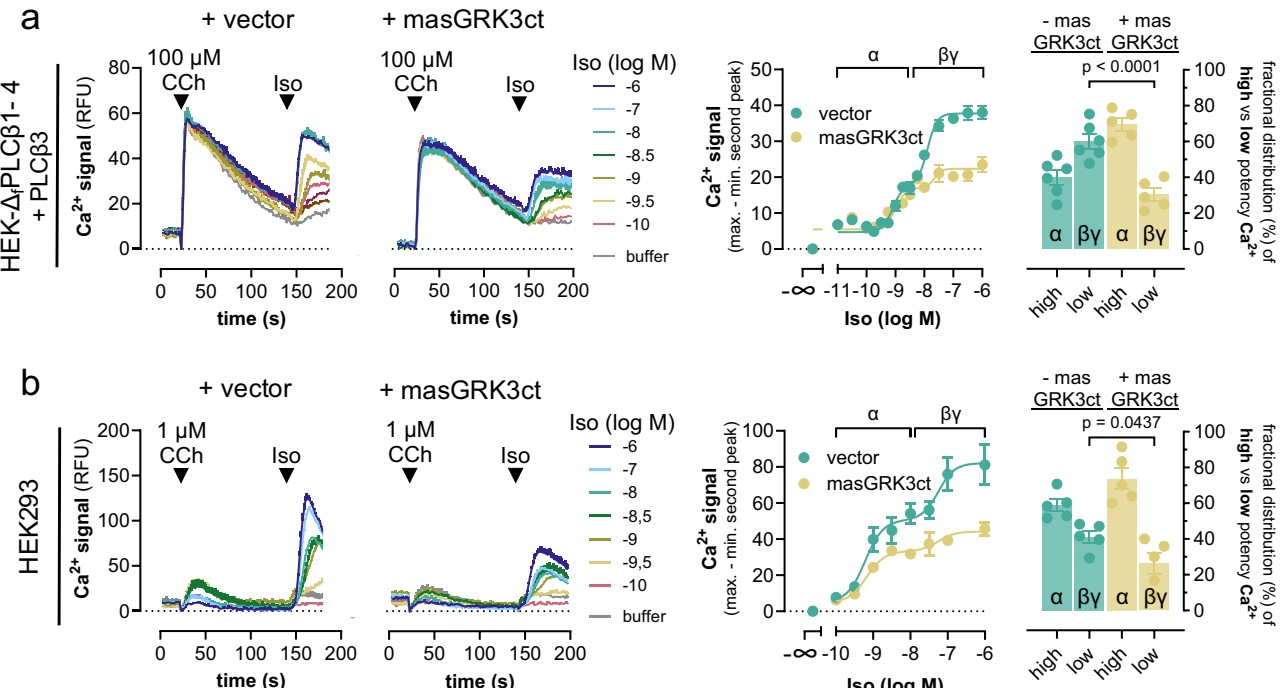

**Fig. 8 | Gs-derived Gβγ drives the Gs-dependent, cAMP-independent Ca²⁺ release pathway. a** Representative Ca²⁺ traces and their quantification evoked in HEK-Δ$_f$PLCβ1–4 cells transiently transfected to re-express PLCβ3 in the absence and presence of the Gβγ-scavenger masGRK3ct. Cells were primed with CCh at $t = 20$ s followed by addition of Iso at $t = 140$ s. **b** Same experimental setup as in (**a**) using HEK293-wt cells and 1 μM CCh as the first stimulus. The concentration-response curves derived from the mean net peak responses are divided into a high-potency

and a low-potency component, reflecting Gα$_s$-cAMP "α" and Gs-βγ "βγ" contribution, respectively. Bar graphs represent the fractional distribution of high- and low-potency Iso-Ca²⁺ and its alteration with co-expressed masGRK3ct. Representative traces are mean + SEM, and data points in concentration-response curves are mean ± SEM of $n$ biologically independent experiments (**a**: vector $n = 6$, masGRK3ct $n = 5$; **b**: $n = 5$), each performed in duplicate. Source data are provided as a Source Data file.

transducers controlling the activity of ACs[83–85], phosphoinositide-3-kinase (PI3)K[86], GPCR kinases 2 and 3[87–89], ion channels[23,90–92], as well as mitogen-activated protein kinase (MAPK) pathways[93–95]. Hence, our findings suggest that Gs-Gβγ signaling, and, potentially, Gβγ freed upon activation of other non-Gi/o proteins, may assume many more functions in cellular signaling than previously anticipated.

## The Gs-Gβγ-Ca²⁺ signaling module extends basic concepts of G protein signal transduction

Gs-Gβγ-Ca²⁺ signaling is remarkable for several reasons. First, it unveils yet another major G protein family that partakes in inositol-lipid signaling suggesting that this form of signal transmission is even more widespread than previously thought. Second, it suggests a hierarchical organization of inositol-lipid signaling with Gq proteins acting upstream of Gs and Gi when Ca²⁺ signals are to be routed through PLCβ proteins. In other words, neither Gi- nor Gs-Gβγ-Ca²⁺ signaling modules act as stand-alone entities but only as part of a Gq-dominated PLCβ-dependent network. In this network, PLCβ3 functions as a bottleneck allowing to pass Gβγ signals onward only if active Gα$_q$ pulls the licensing trigger[32,96]. For PLCβ2, an isoform that is highly abundant in monocytes and neutrophils, chemoattractants and chemokines are known to provoke robust IP₃ and Ca²⁺ responses through Gi-derived Gβγ[74], apparently without coincident Gq activation. We speculate that promiscuous G16 proteins, which are highly abundant in cells from the hematopoietic lineage[97], and which are well known to stimulate PLCβ1–3 isozymes in a manner comparable to that of Gq[98], may assume the licensing function and substitute for Gq in myeloid precursors. Third, although redundant at first glance with Gi-Gβγ-Ca²⁺, Gs-Gβγ-Ca²⁺ is distinct because of enhanced abundance of cAMP within the Ca²⁺ detection window. This cAMP directly increases the sensitivity of ER-localized IP₃Rs to IP₃ and enables detection of two molecularly

separable Ca²⁺ release pathways for the Gβγ-sensitive PLCβ3 only. At low ligand concentrations, Gα$_s$- and cAMP-dependent Ca²⁺ is predominant for all but PLCβ2 isozymes (for the latter likely undetectable due to the low overall signal window after re-expression in PLCβ1–4 KO cells) and mimicked in magnitude by saturating concentrations of forskolin. At higher ligand concentrations, a second cAMP-independent Ca²⁺ release pathway emerges, which is not mimicked by forskolin, but reliant on Gs-derived Gβγ. Precisely this feature, lack of cAMP-dependent Ca²⁺ after Gq priming in primary preACs led us to speculate that the observed Gs-Ca²⁺ is Gβγ-dependent also in this primary cell context.

## Why has Gs-βγ-PLCβ-Ca²⁺ signaling been overlooked for so long?

Possibly because this Ca²⁺ release pathway is obscured by its Gq-dependence and masked by parallel overlapping cAMP-dependent Ca²⁺ release mechanisms in the living cell context. Hence, only if Gq is activated prior to or concomitantly with the Gs-GPCR, and if any cAMP-dependent contribution is disabled, pharmacologically or genetically, Gs-Gβγ-Ca²⁺ will emerge in isolation and be molecularly separable from the remaining Gα$_s$-dependent mechanisms. This explains why Gs-Ca²⁺ may be confounded with Gq-Ca²⁺, because it will always be blunted by FR, regardless of whether Gq input stems from the same Gs-GPCR via secondary Gq coupling or from an independent Gq-GPCR that is co-activated with the Gs-GPCR. It also explains why several parallel pharmacological and genetic perturbations were mandatory to isolate Gs-Gβγ-Ca²⁺: deletion of AC isoforms 3 and 6 to minimize Gα$_s$-dependent cAMP formation, inactivation of PLCβ1–4 alleles to provide a functional zero background for reconstitution of isozyme-specific Ca²⁺ signals, as well as a Gβγ scavenging by the membrane-associated Gβγ effector mimic masGRK3ct. Regardless, the combination of genetic deletion and pharmacological perturbation clearly shows that the

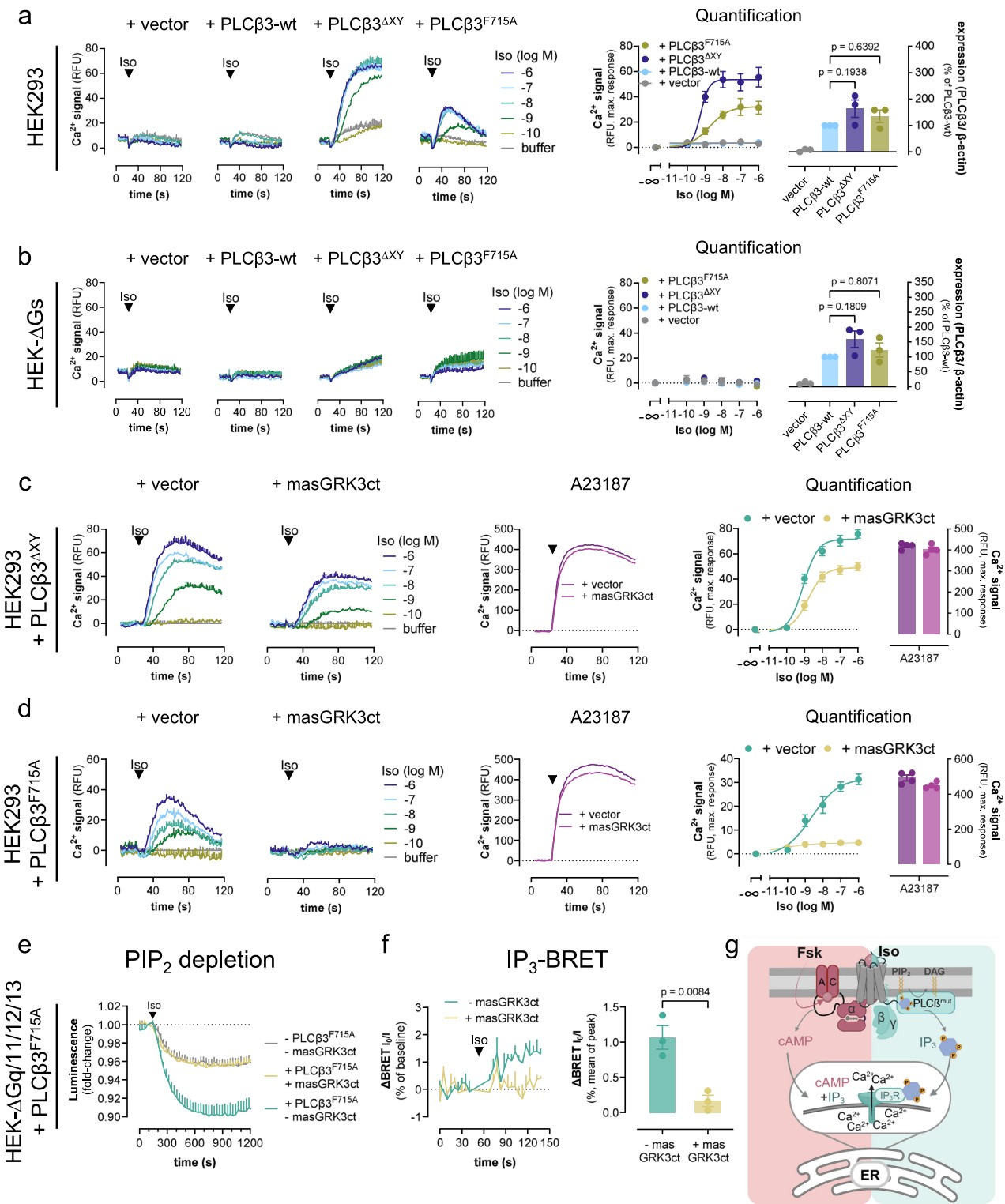

observed pathway is functional in both the recombinant and primary cell environment.

Ca²⁺ signals vary widely depending on the nature of the stimulus and the cellular context. One ubiquitous route to cytosolic Ca²⁺ signals stems from ligand-activated Gq-GPCRs which stimulate PLCβ isozymes to promote PIP₂ hydrolysis into membrane-localized DAG and soluble IP₃. Here we show that PLCβ2/3 isozymes function as coincidence detectors, promoting Gs-GPCR Ca²⁺ only with concomitant or prior action of Gq. Notably, Gs-derived Gβγ but not Gs-α, is the active signaling entity. Thus Gs-GPCRs−via their Gβγ subunits−fine-tune

inositol-lipid signaling to provide a key mediator, Ca²⁺, for mammalian signal transduction. Thereby, they not only expand their signaling versatility but also contribute to one of the most widely used signal transmission processes in eukaryotic cells, previously considered paradigmatic for Gq and Gi-GPCRs only.

## Methods

### Ethics

All animal experiments were performed in agreement with the German law of animal protection and local institutional animal care committees

**Fig. 9 | PLCβ3 variants with disabled autoinhibition empower Iso-mediated Gs-βγ-Ca²⁺ without Gq priming. a, b** Representative Ca²⁺ traces and corresponding quantification of maximum Ca²⁺ amplitudes in HEK293 (**a**) and HEK-ΔGs (**b**) cells transfected with either empty vector DNA or cDNA plasmids coding for PLCβ3-wt, PLCβ3$^{\Delta XY}$ or PLCβ3$^{F715A}$ upon Iso stimulation. Rightmost panels: Western blot quantification of each PLCβ variant. The statistical significance of expression level differences was determined using a one-way ANOVA with Tukey´s post-hoc analysis. **c, d** Naive HEK293 cells were transfected to express either PLCβ3$^{\Delta XY}$ (**c**) or PLCβ3$^{F715A}$ (**d**) in the absence (vector) or presence of the Gβγ scavenger masGRK3ct. **e** Iso-induced PIP₂ depletion in HEK-ΔGq/11/12/13 cells transfected to express the PIP₂ hydrolysis NanoBiT-based biosensor along with PLCβ$^{F715A}$, β₂AR, and masGRK3ct or empty vector DNA as control. **f** IP₃ BRET recordings and corresponding quantification in HEK-ΔGq/11/12/13 cells, transfected to express the IP₃-BRET sensor along with PLCβ$^{F715A}$ in the absence (empty vector) or presence of masGRK3ct upon addition of Iso or buffer. **g** Cartoon representation depicting the cellular consequences of cAMP production as well as IP₃ formation on mobilization of cytosolic Ca²⁺ from ER sources. cAMP and IP₃ synergize to sensitize ER-localized IP₃R channels for mobilization of cytosolic Ca²⁺. Mutant PLCβ3 variants with crippled autoinhibition produce IP₃ without Gq priming in response to Gβγ only. PLCβ$^{mut}$ = PLCβ3$^{\Delta XY}$, or PLCβ3$^{F715A}$. Cells in **a–d** were FR-pretreated (1 μM) to exclude any potential Gq contribution. Representative Ca²⁺ recordings in **a–d** are shown as mean + SEM, summarized data are mean ± SEM of *n* independent biological replicates (**a, b**: *n* = 3; **c, d**: *n* = 4), each performed in duplicate. PIP₂ depletion data (**e**) are mean + SEM of *n* = 3 experiments, each performed in duplicate. BRET IP₃ real-time recordings (**f**) are depicted as mean + SEM of *n* = 3 experiments, one performed in triplicate and two in nonuplicate; their summarized data are shown as mean ± SEM; statistical significance was determined using a two-tailed student's *t* test. Source data are provided as a Source Data file. **g** was created with BioRender.com released under a Creative Commons Attribution-NonCommercial-NoDerivs 4.0 International license https://creativecommons.org/licenses/by-nc-nd/4.0/deed.en".

(Landesamt für Natur, Umwelt und Verbraucherschutz, LANUV). According to the German animal protection law (Tierschutzgesetz) §4 paragraph 3, animals can be sacrificed for scientific purposes/interests for harvesting tissue or isolating cells. Mice were kept in individually ventilated cages in the mouse facility of University Hospital Bonn (Haus für Experimentelle Therapie, Universitätsklinikum, Bonn). Mice were raised under a normal circadian light/dark cycle of each 12 h, and animals were given water and a complete diet (ssniff Spezialdiäten) ad libitum (approved by the Veterinäramt Bonn, §11).

## Reagents and commercial assay kits
Information is provided in Supplementary Table 1 (Reagents) and Supplementary Table 2 (Commercial Assay Kits).

## Cell culture
Parental HEK293 wild-type (wt) and HEK293A cells were obtained from Thermo Fisher Scientific. HEK293-T cells were kindly provided by Jesper M. Mathiesen, University of Copenhagen, Denmark. HEK293 lines lacking G_s and G_olf (ΔGs) or ACs 3 and 6 (ΔAC3/6) were generated using CRISPR/Cas9 technology and characterized as previously described[9,64]. HEK293-wt and HEK-ΔGs cells were cultured in Dulbecco's modified Eagle's medium (DMEM) supplemented with 10% fetal bovine serum (FBS, Sigma), 100 U/ml penicillin and 100 μg/ml streptomycin at 37 °C in a humidified atmosphere of 95% air and 5% CO₂. HEK-ΔAC3/6 cells were cultured in DMEM supplemented with 10% FBS and 1% penicillin–streptomycin-amphotericin B (100×) solution (Thermo Fisher Scientific).

PLCβ1–4-deficient HEK293 cells (HEK-Δ_fPLCβ1–4) were generated from parental HEK293A cells using the CRISPR/Cas9 system. sgRNA constructs targeting the *PLCB1, PLCB2, PLCB3,* and *PLCB4* genes were designed using a CRISPR design tool (crispr.mit.edu) so that a SpCas9-mediated DNA cleavage site (3-bp upstream of the protospacer adjacent motif [PAM] sequence [NGG]) encompasses a restriction enzyme-recognizing site. Designed sgRNA-targeting sequences including the SpCas9 PAM sequences were as follows: 5′-TGTGGGGAA-CATCG**GGCGCCTGG**-3′ (for the *PLCB1* gene; where the BspT107 I restriction enzyme site is underlined and the PAM sequence is in bold), 5′-ACCAGAAACAGCGG**GACTCCCGG**-3′ (for the *PLCB2* gene; Hinf I), 5′-TCATGTCCGTGCTCA**GATCCAGG**-3′ (for the *PLCB3* gene; Mbo I), and 5′-ACAGTTCGGCGGG**AAGTCTTCGG**-3′ (for the *PLCB4* gene; Mbo II). The designed sgRNA-targeting sequences were inserted into the BbsI site of the pSpCas9 (BB)-2A-GFP (PX458) vector (a gift from Feng Zhang at the Broad Institute; Addgene plasmid No. 48138). To generate quadruple PLCB1/2/3/4-mutant cells, we performed a two-step CRISPR/Cas9-mediated mutagenesis with the first round mutating the *PLCB1* and the *PLCB3* genes and the second round mutating the *PLCB2* and the *PLCB4* genes. Briefly, HEK293A cells were seeded into a 10-cm culture dish and incubated for 24 h before transfection. A mixture of the PX458 plasmid encoding the sgRNA and SpCas9-2A-GFP was transfected into the cells using Lipofectamine 2000 (Thermo Fisher Scientific). Three days later, cells were harvested and processed for isolation of GFP-positive cells (~5% of cells) using a fluorescence-activated cell sorter (SH800; Sony). After the expansion of clonal cell colonies with a limiting dilution method, clones were analyzed for mutations in the targeted genes by restriction enzyme digestion. Candidate clones that harbored restriction enzyme-resistant PCR fragments were further assessed for their genomic DNA alterations by direct sequencing or TA cloning. PCR primers to amplify the sgRNA-targeting sites were as follows: 5′-TTTGTGGAATGGGAGCCTTAAAC-3′ and 5′-TGGAAAGCCACGAGATTCAAATG-3′ (*PLCB1*); 5′-GCCCAAGGGA TATGGACCTGTG-3′ and 5′-TGGGGGACAGGAGATAGCTG-3′ (*PLCB2*); 5′-AGTATGAGCCCAACCAGCAG-3′ and 5′-TGAGCAAATGGGCCAAAA GG-3′ (*PLCB3*); 5′-GCCCCAGTCTTCCTAGATCG-3′ and 5′-AAACT-GAAGGGCATCACACAC-3′ (*PLCB4*).

Murine brown pre-adipocytes (preACs) were isolated as previously described[32]. Newborn wild-type C57Bl6/J (Janvier Labs, France) mouse pups were sacrificed and the interscapular brown adipose tissue was harvested and incubated in digestion buffer (DMEM containing 123 mM Na⁺, 5 mM K⁺, 1.3 mM Ca²⁺, 131 mM Cl⁻, 5 mM glucose, 1.5% (w/v) bovine serum albumin, 100 mM Hepes, and 0.2% (w/v) collagenase type II (pH 7.4)) for 30 min at 37 °C. The digested tissue was filtered through a 100 μm nylon mesh to remove cell debris and placed on ice for 30 min. This was followed by further filtration through a 30 μm nylon sieve, followed by centrifugation at 700 × *g* for 10 min. The collected preACs were resuspended in DMEM supplemented with 10% FBS, penicillin (100 U/mL), streptomycin (100 μg/mL), 4 nM insulin, 4 nM tri–iodothyronine, 10 mM HEPES, and sodium ascorbate (25 μg/mL). For immortalization, 60,000 cells per cm² were placed in a 6 cm dish and grown at 37 °C and 5% CO₂. The next day, transduction was performed with lentivirus containing the SV40 large T antigen to immortalize them. Cells were then cultured in DMEM containing 10% FBS, penicillin (100 U/mL), and streptomycin (100 μg/mL) at 37 °C and 5% CO₂ in a humidified incubator as previously described[47].

MEFs were isolated from E13.5 wild-type embryos as described elsewhere[99]. In brief, embryos were collected from a C57Bl6/J (Janvier Labs, France) pregnant mouse, washed with phosphate-buffered saline (PBS) followed by removal of the head, guts, and extremities. After a second round of PBS washing, the embryos were cut into smaller pieces. The tissue fragments were incubated in 0.25% trypsin-EDTA solution at 37 °C for 10 min to allow complete dissociation of the tissue. DMEM supplemented with 10% FBS was added to stop the trypsinization. The embryonic tissue-trypsin-growth medium solution was mixed and transferred into a 50 mL Falcon after filtration through a 100 μm cell strainer. The supernatant was centrifuged at 600 × *g* for 5 min, and the resulting cell pellet was resuspended in DMEM containing 10% FBS, penicillin (100 U/mL), and streptomycin (100 μg/mL) and transferred to a cell culture flask. Cells were then cultured at 37 °C and 5% CO₂, and the medium was changed every 2 to 3 days.

## Plasmids

Gβγ scavenger masGRK3ct inserted into pcDNA3.1 was kindly provided by Nevin A. Lambert, Medical College of Georgia, Augusta[66]. masGRK3ct-Nluc, Venus (156-239)-Gβ1 and Venus (1–155)-Gγ2 were kindly provided by Kirill A. Martemyanov, University of Florida[67]. PLCβ3$^{ΔXY}$ and PLCβ3$^{F715A}$[69] cloned in pcDNA3.1 were kindly provided by John Sondek, North Carolina. For IP$_3$ measurement we used the InsP$_3$R-LBD sensor inserted into pEYFP-C1[65]. The pSNAP-β$_2$AR plasmid was obtained from Cisbio. The pGloSensor™-22 F cAMP construct was from Promega. The muscarinic acetylcholine receptor M$_1$R plasmid was obtained from the Missouri cDNA resource center (Rolla, MO, USA).

The plasmid encoding SmBiT-PLCδ1PH was cloned by exchanging the N-terminal GFP tag in GFP-C1-PLCdelta-PH (Addgene cat.-No. #21179) with a SmBiT-tag (-VTGYRLFEEIL-) via Gibson Assembly. The insert with the sequence for SmBiT was generated by duplexing two complementary oligonucleotides (FW: 5′−AGCGCTACCGGTCGCCAC-CATGGTGACCGGCTACCGGCTGTTCGAGGAGATTCTCTCCGGACTCA-GATCTCGAGCTC-3′; RV: 5′- GAGCTCGAGATCTGAGTCCGGAGAGAAT CTCCTCGAACAGCCGGTAGCCGGTCACCATGGTGGCGACCGGTAGCG CT-3′). In the final expression construct, SmBiT is attached to the N-terminus of the pleckstrin homology (PH) domain of PLCδ1 (1-175). The plasmid encoding the membrane-anchored LgBiT fragment of NanoLuc® (FLAG-LgBiT-CAAX) was generated in the backbone of DEP-Venus-kRas[100]. The nucleotide sequence for LgBiT was amplified from LgBiT-β-arrestin2[101] (note that a N-terminal FLAG tag was attached during the PCR) and the PCR product was used to replace DEP-Venus in DEP-Venus-kRas via Gibson Assembly.

All other plasmids were generated by PCR and inserted into pcDNA3.1. Sequences of newly generated plasmids were verified by Sanger Sequencing (Eurofins Genomics).

## Ca$^{2+}$ mobilization assay

Intracellular calcium measurements were conducted using the FLIPR Calcium-5 Assay Kit (Molecular Devices, San Jose, CA, USA), which contains the Calcium-5 dye as well as masking dyes to reduce background fluorescence and improve the signal-to-noise ratio, according to the manufacturer's protocol. In brief, for HEK and MEF cells, 60,000 and 50,000 cells were respectively seeded per well into Poly-D-lysine (PDL)-coated 96-well plates and cultured overnight. The next day, the medium was removed, and each well was filled with 50 μL of Calcium-5 indicator, which previously was reconstituted in Hanks´s Buffered Saline Solution (HBSS) containing 20 mM HEPES (HBSS + HEPES), and incubated at 37 °C and 5 % CO$_2$ for 45 min. Afterwards, either 150 μL, for the single ligand addition Ca$^{2+}$ assay, or 100 μL, for the double ligand addition Ca$^{2+}$ assay, of HBSS + HEPES were added to the wells and incubated for 15 min at 28 °C before starting the measurement. Ca$^{2+}$ mobilization was determined as increments in fluorescence intensity over time acquired by the FlexStation® 3 Multimode Bench Top reader (Molecular Devices, San Jose, CA, USA), which expresses them as relative fluorescence units (RFU). The first 20 s of the measurement served as the initial baseline read. Subsequently, 50 μL of the compound was added either once after 20 s or twice at 20 s and 140 s. The measured fluorescence counts were normalized according to their initial baselines and then adjusted to the first compound addition, which was set to $y = 0$ as previously described[32]. A23187 (5 μM) was used as a receptor-independent stimulus to increase cytosolic Ca$^{2+}$. For Gi/o inhibition analyses, cells were pre-incubated with 100 ng/mL PTX for at least 16 h. Signaling pathway inhibitors FR (1 μM), PKI14-22 (10 μM), and HJC0197 (25 μM), to block Gq, PKA, and EPAC, respectively, were mixed directly into the Calcium-5 dye and incubated for 1 h before starting the measurement. IP$_3$R inhibitor 2-APB (50 μM for HEK293 cells and 100 μM for ΔAC3/6 cells), diluted in HBSS + HEPES, was added with a short incubation time of 15 min at 28 °C, which was preceded by a longer incubation time of 45 min with the Calcium-5 dye before starting the measurement. For preACs, cells were seeded into PDL-treated 96-well plates at 16,000 cells/well and then cultured at 37 °C for 48 h; 100 ng/mL PTX was added 16 h before the measurement. Final quantification was performed with buffer-corrected values, processed as described in the $y$ axis labels of the graphs.

For Ca$^{2+}$ assays with transfected HEK293 cell lines, linear polyethylenimine (PEI, 25 kDa, Polyscience) was used 48 h before measurement, as previously described[102]. As a rule, 2.5 μg total DNA and 7.5 μL PEI solution (1 mg/mL) were used for $1 × 10^6$ (DNA:PEI ratio of 1:3). When necessary, the DNA mixture was supplemented with an empty vector to achieve the final amount of DNA. For analyses of overexpressed β$_2$AR, 1.675 μg of the receptor with or without 0.825 μg of Gα$_q$ was used per million cells. For re-expression of PLCβ1−4 isoforms in HEK-Δ$_f$PLCβ1−4 cells, we transfected 2 μg PLCβ1, 0.5 μg PLCβ2, 2 μg PLCβ3, and 2 μg PLCβ4, complemented by empty expression vector to 5 μg of total DNA, into 2 million cells. For Gβγ scavenging analysis, 4 μg of masGRK3ct were transfected in HEK293 cells, or 0.25 μg of masGRK3ct together with 2 μg of PLCβ3 in HEK-Δ$_f$PLCβ1−4 cells (2 million cells, 5 μg total DNA). For examination of the PLCβ3 variants, 2 μg of PLCβ3-wt, PLCβ3$^{ΔXY}$, or PLCβ3$^{F715A}$ were transfected in HEK293 and HEK-ΔGs, and for Gβγ scavenging analysis, 4 μg of masGRK3ct were transfected together with 1 μg of PLCβ3$^{ΔXY}$ or PLCβ3$^{F715A}$ in HEK293 (2 million cells, 5 μg total DNA).

## cAMP accumulation assay

For cAMP measurements, we used the HTRF-cAMP dynamic 2 kit (Cisbio Codolet, France) and a suspension cell-based protocol. In brief, HEK cells were detached, resuspended in a stimulation buffer (HBSS + HEPES), and seeded (5000 HEK cells/well) into a white 384-well microtiter plate. The cells were allowed to equilibrate in the plate for 20 min at 37 °C and then stimulated with increasing concentrations of receptor agonists for 30 min. The assay was terminated by the sequential addition of d2-labeled cAMP and cryptate-labeled anti-cAMP antibody, then leaving the plate in the dark for at least 1 h at room temperature. To record HTRF values, we used the Mithras LB 940 multimode plate reader (Berthold Technologies, Bad Wildbad, Germany) at 665 and 620 nm. Using a standard curve generated from the cAMP standard solutions provided by the manufacturer, all HTRF ratios were converted to nM cAMP concentrations.

## Real-time GloSensor™cAMP detection assay (population-averaged cAMP determination)

Ligand-mediated dynamic changes of intracellular cAMP levels were monitored using the GloSensor™cAMP biosensor (Promega Corporation, Wisconsin, USA) according to the manufacturer's instructions. In brief, 0.8 million HEK cells were transfected with 1.5 μg of pGloSensor™-22F cAMP plasmid with PEI (1 mg/ml, 1:3 ratio) in a 6 cm dish. 24 h post transfection, the cells were harvested, washed with PBS, centrifuged, and resuspended with HBSS + HEPES. 50 μL of 50,000 cells/well were seeded into a flat-bottomed 96-well plate followed by the addition of 50 μL of GloSensor™ cAMP substrate (2%) in HBSS + HEPES. The cells were then incubated for 2 h at room temperature. The PHERAstar microplate reader (BMG labtech, Ortenberg, Germany) was used to measure cAMP-BRET at an emission wavelength of 562 nm, after two additions of 50 μL compounds at 20 s and 120 s. To detect real-time cAMP formation, we collected technical triplicates for each compound with a 3 second acquisition time per data point. The first 20 s were used as a baseline. The luminescence signals were normalized to the solvent-primed Iso addition.

## Real-time, FRET-based, single-cell cAMP detection assay

Live-cell cAMP measurements in MEFs and preACs were performed using the pcDNA3.1-mICNDB-FRET sensor[103]. MEFs and preACs were transfected using the SF Cell Line 4D-Nucleofector X Kit (Lonza). In

brief, 1,000,000 cells of each cell line were spun down at $360 \times g$ for 4 min at RT. Cells were resuspended in 100 µl of nucleofection master mix, consisting of 82 µl SF Solution, 18 µl of Supplement 1, and 2.5 µg of mICNBD-FRET plasmid DNA. Cell suspensions were transferred into separate Nucleocuvettes and electroporated in the 4D-Nucleofector X Unit (Lonza) with the pulse codes CZ 167 (MEFs) and CA 158 (preACs). After electroporation, the cell suspensions were incubated at RT for 10 min before the addition of 1 ml of pre-warmed media. The cell suspensions were then seeded at a density of 80,000 cells per well of a black PhenoPlate 96-well plate (Revvity) and incubated at 37 °C and 5% $CO_2$ for 24 h before the measurement. For cAMP measurements, the cells were washed once with 200 µl per well of extracellular solution (ES: 10 mM HEPES pH 7.4, 120 mM NaCl, 5 mM KCl, 2 mM $MgCl_2$, 2 mM $CaCl_2$, 10 mM glucose), followed by addition of 180 µl of ES into each well. Imaging was performed using the Zeiss Axio Observer Z1 with a dual camera setup operating at 37 °C. FRET was recorded by exciting cerulean at 436 nm and measuring the emission of cerulean and citrine at 470 nm and 535 nm, respectively. The measurements were performed at 10 s intervals, with the first 6 images (60 s) serving as the baseline, followed by the addition of the stimulants at the indicated timepoints. To measure the cAMP increase in response to Iso without prior Gq priming, cells were stimulated with a final concentration of 10 µM (MEFs) or 1 µM (preACs) Iso. For Forskolin, a final concentration of 30 µM was used for both cell lines. Gq priming was performed with 1 µM ATP (MEFs) or 10 µM 5-HT (preACs) after 60 s, followed by addition of Isoproterenol or Forskolin after 180 s. The change in FRET was calculated using the sensitized emission (SE) of citrine and cerulean ($SE_{Citrine} - E_{Cerulean}/E_{Cerulean}$). Data were collected using Microsoft Excel and analyzed with GraphPad Prism software. The first 6 images were used as mean baseline. The inverse of the difference (value−baseline/baseline)*100 was calculated at each time point (ΔFRET values). Corrected ΔFRET values at each time point were determined by subtracting the mean ΔFRET values obtained at the same time point in vehicle-treated wells. Data are presented as mean + SEM.

## Gq-CASE BRET assay

BRET measurements were performed using the Gq-CASE biosensor as previously described[52]. 300,000 resuspended HEK293 cells/mL were transfected with a 1:1 ratio of $\beta_2AR$ and Gq-CASE using 3 µL PEI solution per µg total DNA and seeded directly onto PDL-coated white 96-well plates (30,000 cells per well). At 48 h post transfection, cells were washed once with 120 µL HBSS per well and incubated with a 1/1000 dilution of NanoBRET™ NanoGlo® substrate for 2 min at 37 °C. After three baseline BRET readings, serial dilutions of isoproterenol or HBSS (buffer) were added to the cells and BRET was recorded for ten consecutive readings. BRET measurements were performed using the CLARIOstar Plus multimode plate reader (BMG labtech, Ortenberg, Germany). Emission at $470 \pm 40$ nm and $530 \pm 15$ nm was measured with an integration time of 0.3 s, a focal height of 10 mm, and gain settings of 3000 and 3600, respectively. The data were buffer-corrected and quantified by determining the mean BRET decrease after addition of the substances as previously described[52]. Briefly, the raw ΔBRET (%) over the three baseline measurements at each time point t was calculated as (($BRET_t$−mean baseline BRET)/mean baseline BRET) × 100. The corrected ΔBRET values at each time point were determined by correcting for vehicle-induced changes in BRET, i.e., by subtracting the mean raw ΔBRET values obtained at the same time point in vehicle-treated wells.

## IP$_3$ BRET assay

BRET measurements were performed using the IP$_3$ sensor[65] as previously described[32]. HEK293-wt (1.2 million cells) were transfected with 300 ng of the IP$_3$ sensor. HEK ΔGq/11/12/13 (1.2 million cells) were transfected with 200 ng of $\beta_2AR$, 1 µg of PLCβ3$^{F715A}$ and 20 ng of the IP$_3$

sensor, alone or together with 800 ng of masGRK3ct. The total DNA amount used in the transfection was 3 µg using PEI (DNA:PEI ratio of 1:3) and carried out in 6 cm dishes 48 h before the measurement. On the day of the assay, cells were trypsinized and washed twice with PBS. The cell pellet was then resuspended in HBSS + HEPES to seed 80 µL of 80,000 cells/well into a flat white-bottom 96-well plate (Corning). After the addition of 10 µL BRET substrate Coelenterazine h, the plate was briefly placed on a shaker to allow uniform distribution of the substrate. BRET measurements were carried out with the PHERAstar FSX multimode plate reader (BMG Labtech, Ortenberg, Germany). Emission at 485 nm and 535 nm was measured for 50 s as the initial baseline read before 10 µL compound addition. In two-compound-addition assays, the second addition occurred 120 s after the first compound, similar to the calcium assays. BRET ratios were normalized dividing the mean of the baseline BRET ($I_0$) by those at each time point ($I$). The raw ΔBRET (%) at each time point was calculated by subtracting the normalized baseline BRET values and multiplying by 100. The corrected ΔBRET values at each time point were determined by subtracting the mean raw ΔBRET values obtained at the same time point in vehicle-treated wells.

## Free-G$_{\beta\gamma}$-GRK-BRET-assay

Gq activation of exogenous $\beta_2AR$ was measured with the Free-G$_{\beta\gamma}$-GRK-BRET-Sensor[67]. 350,000 HEK293-T cells were seeded into a 6-well plate and transfected after 24 h using Polyethyleneimine (PEI, 1 mg/ml) in a ratio of 1:3 (DNA:PEI) and cDNA plasmids in the following amounts per dish: masGRK3ct-Nluc (0.025 µg), Venus (156-239)-Gβ1 (0.2 µg), Venus (1-155)-Gγ2 (0.2 µg), murine Gαq (1 µg), $\beta_2AR$ (0.4 µg). Empty pcDNA3.1 was added to a total amount of 2 µg DNA per well. 27 h after the transfection, the cells were washed with PBS, detached by scraping, centrifuged at $500 \times g$ for 3 min, and resuspended in HBSS buffer supplemented with 20 mM HEPES. Approximately 20,000 cells per well were added into a white, flat-bottom 96-well plate (Corning, USA) and pre-incubated with FR (10 µM) or vehicle (DMSO) for 15 min. After adding NanoGlo® Luciferase substrate (Promega) up to a final dilution of 1:1,000, luminescence (475 nm±15) and fluorescence (535 nm±15) measurements were performed at 28 °C in the PHERAstar® FSX plate reader (BMG labtech, Ortenberg, Germany) with a 1.44 s measurement interval time for 300 s. Iso (1 µM) was added after a baseline read of 90 s. The BRET ratio was calculated by dividing fluorescence emission by luminescence emission values. It is shown as the mean + SEM of 3 independent experiments as buffer-corrected BRET ratios (ΔBRET) generated by subtracting the last baseline read before the addition. For statistical analysis, the mean of three experiments was pooled after reaching the plateau, 200 s after the start of the measurement. Two-way analysis of variance (ANOVA) test was performed to analyse these values.

## HTRF-based IP$_1$ accumulation assay

IP$_1$ accumulation was measured using Revitys' HTRF IP-One Gq Detection Kit, according to the manufacturer's protocol. In brief, 1.2 million HEK293 cells were transfected with $\beta_2AR$ (2500-3000 ng) and pcDNA3.1 up to a total amount of 3000 ng using Polyethyleneimine (PEI, 1 mg/ml) in a ratio of 1:3 (DNA:PEI) in 6-cm dishes. After 48 h, the cells were detached, washed with PBS, and then resuspended in LiCl-containing stimulation buffer to prevent IP$_1$ degradation. To determine the Gq contribution, FR (10 µM) or vehicle (DMSO) was added to the stimulation buffer, and the cells were incubated for 15 min. To investigate ligand-induced IP$_1$ accumulation, 35,000 cells in suspension were transferred into a 384-well plate together with Iso (1 µM), CCh (100 µM) or buffer and incubated for 30 min before the addition of the IP$_1$ d2 Reagent, and the IP$_1$ Tb Cryptate antibody together with lysis buffer. After 60 min, HTRF ratios were measured on a Mithras LB 940 reader (Berthold Technologies, Bad Wildbad, Germany) according to the manufacturer's instructions. HTRF ratios were converted into

nanomolar concentrations of $IP_1$ according to the kit's standard curve using nonlinear regression analysis. The statistical analysis of the data was performed using a two-way ANOVA test.

## NanoBiT-based plasma membrane $PIP_2$ depletion assay

The rate of plasma membrane $PIP_2$ depletion was measured using NanoLuc® binary technology (NanoBiT). This system is composed of two NanoLuc® subunits, a smaller fragment SmBiT and a larger fragment LgBiT fused to target proteins of interest. Here, we adapted this system to generate a NanoBiT-based biosensor for the quantification of plasma membrane $PIP_2$ depletion. We fused the isolated PH domain of phospholipase C$\delta$1, a bona-fide $PIP_2$ binding protein frequently used as GFP fusion to visualize phosphoinositide dynamics[104–106], to the small fragment of NanoLuc® luciferase (SmBiT-PLC$\delta$1PH) and generated a membrane-anchored version of the larger NanoLuc® fragment (LgBiT-CAAX). We confirmed plasma membrane localization by confocal imaging (Supplementary Fig. 20a) and functionality of the biosensor using stimulation of the Gq-coupled muscarinic M1 receptor (M$_1$R) in HEK293A (Supplementary Fig. 20b–e) and our HEK-PLC$\beta$1–4 knockout cells (Supplementary Fig. 20f, g).

NanoBiT assays were conceived by transfecting HEK293A cells in suspension (350,000 cells/mL) with 100 ng of SmBiT-PLC$\delta$1PH, 200 ng of M$_1$R, and 250 ng of FLAG-LgBiT-CAAX; HEK-$\Delta_f$PLC$\beta$1–4 with 100 ng of SmBiT-PLC$\delta$1PH, 200 ng of M$_1$R, 250 ng of FLAG-LgBiT-CAAX and 200 ng PLC$\beta$3; HEK $\Delta$Gq/11/12/13 cells with 50 ng of SmBiT-PLC$\delta$1PH, 125 ng of FLAG-LgBiT-CAAX, 100 ng of $\beta_2$AR, 300 ng of PLC$\beta$3$^{F715A}$ and 425 ng of masGRK3ct. Empty pcDNA3.1 was used to adjust the final DNA amount to 1 µg total DNA per mL cell suspension and transfected using PEI, at a DNA:PEI ratio of 1:3. The transfected cell suspension was then transferred to a PDL-coated, white 96-well plate (35,000 cells/well).

After two days of incubation at 37 °C (5% $CO_2$), cells were washed once with HBSS and incubated with 80 µL of HBSS in the presence or absence of 1 µM FR for 1 h at 37 °C. Next, 10 µL of furimazine (Promega, #N1120, final dilution 1/1000) were added and the plate was incubated for 15 min inside the plate reader (pre-warmed to 37 °C) before the measurement was started. After five baseline reads, 10 µL of the vehicle or CCh was added, and the measurement was continued. For priming experiments, a second compound addition was performed after seven minutes. All experiments were performed at 37 °C using a TECAN Spark multimode plate reader. Bioluminescence originating from the complemented luciferase was detected between 460 and 500 nm with an integration time of 0.1 s. Raw data were baseline-corrected (correction for baseline drift), buffer-corrected (for priming experiments: corrected for "solvent prime + vehicle") and plotted as "fold-change in luminescence".

## Confocal microscopy

The subcellular localization of FLAG-LgBiT-CAAX was assessed via confocal microscopy. HEK293A cells were seeded on PDL-coated four chamber 35 mm dishes (Ibidi, #80416) and transfected with either FLAG-LgBiT-CAAX using the same plasmid cDNA amount as in the NanoBiT complementation assays (see above) or empty pcDNA3.1 as negative control using PEI as the transfection reagent. After fixation with 4% paraformaldehyde and permeabilization, immunostaining of the N-terminal FLAG tag was performed following a previously described protocol[107]. The anti-FLAG M2 antibody (Sigma Aldrich, #F1804, 1:1000) and the polyclonal goat anti-mouse Alexa Fluor 488-conjugated antibody (Invitrogen, #A28175, 1:1000) were used as the primary and secondary antibody, respectively. Nuclei were counterstained with 1 µg/mL Hoechst 33342 for 5 min. Confocal images were recorded using a Zeiss LSM800 confocal microscope.

## Label-free dynamic mass redistribution (DMR) assay

DMR studies were performed using the Corning® EPIC® biosensor (Corning, NY, USA), according to previously published protocols[108].

18,000 HEK cells/well were seeded into a 384-well biosensor plate (Corning NY, USA) and cultured overnight. On the day of measurement, cells were washed with HBSS + HEPES and equilibrated for 1 h in the DMR reader at 37 °C. FR (1 µM) or 2-APB (50 µM) were added 1 h before the measurement in HBSS + HEPES. The sensor plate was scanned to record a baseline optical signature for 5 min, and then compounds were added using the CyBio SELMA semi-automated electronic pipetting system (Analytik Jena AG, Jena, Germany). After addition, measurements were recorded for 60 min at 37 °C. Ligand-mediated changes in DMR were quantified as the pm shift in reflected wavelength over time.

## Western blot

Protein samples prepared from cells transfected with wild-type or mutant PLC$\beta$3 variants for calcium determinations were separated by 10% SDS-polyacrylamide gel electrophoresis and transferred onto nitrocellulose membranes by electroblotting. Nitrocellulose membranes were washed, incubated with ROTI®Block (1×; Carl Roth) for 1 h and incubated overnight at 4 °C in ROTI®Block with anti-$\beta$-actin antibody (BioLegend, #622102, 1:10,000). Afterward, membranes were washed and incubated for 1 h at room temperature with a horseradish peroxidase-conjugated secondary antibody (goat anti-rabbit IgG Antibody HRP; ABIN, #102010, 1:20,000). The $\beta$-actin band was detected using the Amersham Biosciences ECL Prime Western blotting detection reagent (GE Healthcare). Membranes were stripped and reprobed in ROTI®Block with an antibody against the anti-PLC$\beta$3 mouse monoclonal antibody (Santa Cruz Biotechnology, sc-133231, 1:500) and incubated overnight at 4 °C. Anti-mouse antibody (goat anti-mouse IgG antibody HRP; Sigma, #A4416, 1:20,000) diluted in Roti®Block was used as a second antibody for detection.

## Surface protein expression quantification assay

Cell-surface $\beta_2$AR levels were quantified after transfection of SNAP-tagged $\beta_2$AR as follows. 24 h after transient transfection, cells were washed, trypsinized, and 60,000 cells/well were transferred to a PDL-coated 96-well plate and cultured for additional 24 h. Then, medium was removed and 50 µL/well of 100 nM SNAP-Lumi4-Tb reagent diluted in HBSS + HEPES were added to the cells and incubated for 1 h at 4 °C. Afterwards, the SNAP-Lumi4-Tb reagent was removed and 100 µL of HBSS was added to each well. The PHERAstar FSX multimode plate reader (BMG labtech, Ortenberg, Germany) was used and emission was measured as RFU at 620 nm after excitation at 337 nm. Surface protein quantification was performed in parallel with the same batch of transfected cells used in the $Ca^{2+}$ assay.

## Data and statistical analysis

Data were collected with Microsoft Excel 2019 and analyzed using GraphPad Prism 10.2.3 software. Concentration-effect curves were fitted to four-parameter logistic equations if satisfactorily described by the classical Hill equation. Concentration-effect curves with more than one inflection point were fitted with biphasic equations applying constraints when fits were ambiguous. Representative data are displayed as mean + SEM, and quantified data as mean ± SEM. Some of the kinetic recordings are presented as buffer-corrected traces by subtracting the corresponding buffer-stimulated data point from the ligand-stimulated data point at each measured time. ANOVA, or two-tailed Student's $t$ test was used for statistical analysis as indicated in the figure legends and text.

## Reporting summary

Further information on research design is available in the Nature Portfolio Reporting Summary linked to this article.

# Data availability

The data that support this study are available from the corresponding author upon request. All data generated and analyzed during this study

are included in this published article and the Supplementary Information. Source data are provided with this paper.

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

## Acknowledgements

Funded by the Deutsche Forschungsgemeinschaft (DFG, German Research Foundation) with the grants 214362475/GRK1873/3, 290847012/FOR2372, and 494832089/GRK2873 to E.K., 542889291/Emmy Noether fellowship to H.S., 390873048/EXC2151 to D.W., 450149205/TRR333/1 to D.W. and A.P., 504098926 to L.G., and by the Swedish Research Council with grant 2019-01190 to G.S. J.B. and L.J. were members of the DFG-funded Research Training Groups RTG1873 (214362475/GRK1873/3) and RTG2873 (494832089/GRK2873), respectively. A.I. was supported by KAKENHI JP21H04791, JP21H05113, JP21H05037, and JPJSBP120213501 from the Japan Society for the Promotion of Science (JSPS); JP22ama121038 and JP22zf0127007 from the Japan Agency for Medical Research and Development (AMED); JPMJFR215T, JPMJMS2023 and 22714181 from the Japan Science and Technology Agency (JST). We gratefully acknowledge the technical assistance of Ulrike Rick and Tania Gross.

## Author contributions

J.B., S.B., L.J., L.G., H.S., F.F., J.A., C.P., P.G.O., S.H., L.S., N.H., and K.K. conceived and performed experiments, and analyzed the data; S.H., K.K., and C.P. generated the PLCβ1–4 CRISPR knockout line, supervised by A.I.; V.J.W, A.P., L.S., T.S., and G.M.K provided valuable cells or reagents; A.I., G.S., D.W., and V.J.W contributed valuable advice and discussion; J.B., S.B., C.P., K.S., and E.K. conceived and designed the study; A.I., K.S. and E.K. coordinated, supervised and oversaw the study; E.K. and J.G. take responsibility for the data integrity and accuracy of data analysis; C.P. drafted the initial version of the manuscript; K.S. and J.B. drafted the second version; E.K. wrote the manuscript with input from and editing by all authors.

## Funding

## Competing interests

The authors declare no competing interests.
