## [Peer Review File · Nature Communications]

A molecular mechanism to diversify Ca²⁺ signaling downstream of Gs protein-coupled receptorsREVIEWER COMMENTS

Reviewer #1 (Remarks to the Author):

In this manuscript, Brands et al propose a novel mechanism of Ca²⁺ mobilisation dependent on Gβγ released from the association with Gs. The authors use a calcium mobilisation assay to comprehensively evaluate calcium signalling over a wide number of cellular conditions, including endogenously expressing cells and HEK293 cells with KO of various signalling partners. It is concluded that following Gq priming, Gs-coupled GPCR activation produces activated free Gβγ complexes alter inositol lipid signalling to cause an increase in Ca²⁺ signalling. This new mechanism extends our understanding of the canonical GPCR-calcium signalling and is important for the field.

Major comments:

1. In Figure 1, the authors use PTX to block the native Gi/o activation for panel C and D, but not for β2AR in panel B. As β2AR could also couple to Gi/o, it would be more convincing to collect data for β2AR in presence of PTX.
2. Figure 3, panel C claims that exogenous β2AR can couple to Gq which requires further support as it contradicts prior observations. The maximum BRET signal change reported in this figure is only ~2% of the baseline, which does not help instilling the confidence in the conclusions either. There are several well-established biosensor assays that could detect G protein coupling. It would be preferred to use unmodified Gq in these assays to avoid artifactual coupling. It would be also nice to see the response of an agonist at a canonical Gq-coupled receptor to compare whether the small BRET ratio reported indicates physiological significance.
3. In figure 4E, the calcium signal of 1 uM Iso is about 170 RFU. however, the calcium signal for 1 uM Iso is about 200 RFU in figure 1A and 1B and is 80 RFU for w/o in figure 5A. These experiments were done under exact same conditions. What explains these differences which exceed variability shown by error bars?
4. Concentration response curves in Figure 4G-H could be fit with a singular sigmoidal curve instead of the “biphasic” curve. More explanation should be given regarding why the “biphasic” curve was used, when it is not apparent from the data. A curve fit with two inflection points is likely to fit data better, as this allows for all data points to be on the curve fit, compared to a larger number of data points that fall outside of the sigmoidal curve fit usually used for this type of data. The “biphasic” curve was found to be integral in determining the mechanism of action, however the reasoning for this curve fitting is not justified. A more elaborate description of what analysis was performed and what constraints were used in the methods section would also help guide the reader.
5. In figure 4F, iso induced lower maximum calcium signal in HEK-ΔAC3/6 cell; and figure 5A shows that IP3R inhibitor 2-APB could eliminate Iso-mediated β2AR-calcium signal after Gq priming. It would be important to apply 2-APB to HEK-ΔAC3/6 cell to see whether it could eliminate Iso-mediated β2AR-calcium signal, which could strengthen the conclusion that cAMP-independent pathway is IP3R-dependent.
6. BRET data showing IP3 formation in Figure 5C (and also Figure 8E) is also not convincing. From the bar graph in Figure 5C it appears that the difference between buffer (approx. 0.006) and 1 μM Iso (approx. 0.01) stimulation is approximately 0.004. This is a very small

Δ BRET ratio therefore, this experiment would also benefit from having a positive control, to determine whether this small change indicates physiological relevance. Alternatively, a more robust IP3 formation assay should be used, as the conclusion from this experiment is instrumental to lead to Figures 6-8 in the manuscript.

7. The case for the exploration and the investigations performed are quite clear but the writing style is very verbose with an abundance of unnecessary complex arguments, flashy statements and stump words. It is advisable for authors to clean it up- make the Introduction concise and to the point, avoid restating motivation and logic in Results and focus them better on design and data analysis and streamline the Discussion section as well. The readers will appreciate it as the key message of the manuscript is quite simple at the end of the day.

Minor comments:

1. All abbreviations used should be defined where first mentioned.
2. Line 32 and line 524 should be "A long-held tenet in inositol lipid signaling that cleavage...".
3. The latter part of the sentence on lines 100-103 should be simplified, as it is difficult to determine the meaning of this sentence.
4. Line 277 could be written in more concise way, e.g. "The mechanism of Gq coordinates"
5. Justification for why the specific isoforms of AC (AC3 and AC6) were deleted should be provided.
6. The term 'biphasic' should be changed, as there are not two phases occurring at different times, rather two modes of action at different concentrations of Iso.
7. Why was a two-way ANOVA used to assess changes in maximal effect of Iso in the absence and presence of PKI14-22 (Figure 4A)?
8. In Figure 6 (quantification panel) the concentration response curves of Ca signal in panel B and C are in series with the vehicle point, however in panel D and E the vehicle condition is not in series. What is the reasoning for this? And also, why in panel D and E the bottom of the curve begins at a Ca²⁺ signal of 5 and not at 0.
9. Reference to Figure 7C, D on line 472 should in fact reference Figure 8.
10. Please consider rephrasing sentence on line 554-555 for clarity, e.g. "proves the physiological significance of Gs-G $\beta\gamma$ -Ca²⁺ signaling".
11. Sentence on line 564-568 is very long and hard to understand "Third, although redundant...PLC β 2 and β 3 isozymes only.
12. There is only two data points on the bar graph in Figure 8B and no error is shown, however in the figure caption it states that all data is from three independent biological experiments. This figure should be updated to include a third dataset.
13. ATCC number should be included in the methods section, as on line 638 it states that HEK293 wild-type cells were used, however on line 651 it states that HEK293A cells were used. Clarity around which cell line was used is needed.
14. More explanation describing the method to determine the "mean BRET decrease" for BRET experiments should be provided in the methods section.

Reviewer #2 (Remarks to the Author):

This is an important, elegant, and well presented study. The manuscript is a fantastic example of how to carry out a systematic and rigorous study of mechanisms of signal transduction. The focus is on a previously unknown mechanism by which GPCRs mobilize intracellular calcium using G $\beta\gamma$ released by from GPCR-stimulated Gs heterotrimers. The study reveals the specific mechanistic details how two of the most prevalent second messenger systems (dependent on cAMP or on Ca²⁺) intersect with each other in yet another unexpected manner by using a combination of genetic and pharmacologic manipulations. The conclusions are well supported by the data, which is presented beautifully (I even like choice of color palettes). The work is systematic and comprehensive, and does an excellent job at dissecting different components of the signaling response to Gs-coupled GPCRs, including confirmatory studies in more than the frequently used cell line HEK293. The introduction and discussion sections are very clear, despite the complexity of the topic an extensive background. In summary, I am glad to recommend to publish this work as is without reservations.

Reviewer #3 (Remarks to the Author):

This is an interesting and well controlled series of experiments examining how Gs coupled G protein coupled receptors lead to increased Ca²⁺ mobilization through obligate crosstalk with G α_q coupled receptors. Evidence is presented showing that G $\beta\gamma$ released from Gs coupled receptors synergizes with G α_q released from G α_q coupled receptors at the level of phospholipase C beta activation to increase PLC activity leading to enhanced IP₃ production and Ca²⁺ release. This follows on previous work from these investigators demonstrating obligate G α_q priming of G $\beta\gamma$ signaling from G α_i coupled receptors, which in turn follows on earlier work demonstrating that activation of PLC β_3 by G α_q and G $\beta\gamma$ subunits is strongly synergistic providing a mechanistic basis for G α_q priming.

The authors characterize a dual mechanism for receptor crosstalk and Ca²⁺ mobilization: at low Gs-GPCR agonist concentration cAMP sensitizes the IP₃ receptor to IP₃; at higher agonist concentration G $\beta\gamma$ -G α_q PLC activation predominates. These results are significant in that they move beyond the prevailing view that G $\beta\gamma$ signaling to PLC and other targets derives solely from G α_i coupled receptors and suggests that G $\beta\gamma$ released from G protein heterotrimers other than G α_i can participate in downstream signaling processes. The overall experiments are very thorough but there are some issues that should be addressed.

1) The major concern that I have is with the dissection of the two modes of Ca²⁺ release sensitization by Gs coupled receptors that is proposed. As indicated by the authors regulation of Ca²⁺ release is regulated by many factors other than simply phospholipase C activation and Ca²⁺ release is an indirect measure. Thus, the analysis must be conducted very carefully. While the experiments measuring Ca²⁺ are somewhat convincing, to really show synergy at the PLC level, measurements of the PLC reaction products, either IP, IP₃ or DAG would be much more definitive and should applied to a greater extent and with greater rigor than is shown here. To the authors credit they tried to do this using a BRET based IP₃ reporter. While suggestive, I find these figures 5 and 8E less than convincing. There is a small but significant increase with ATP and the addition of Iso marginally increases this. More

convincing would be to prime with a concentration of ATP that gives a very small IP3 response on its own, which should in turn lead to a more pronounced increase in IP3 when Iso is added. This should work because 3 μ M ATP is able to prime the iso response. That being said an independent method that does not rely on BRET directly measuring IP accumulation, IP3 production or DAG should be used to more definitively make the case.

To this point, the recognition that cAMP sensitization of IP3 receptors is one mechanism for Gq-Gs crosstalk is carefully considered here but is indirect. The authors claim that the dose response curves in figures 4G and H, and 6 D and E are biphasic, with cAMP-IP3 receptor sensitization operating at low ligand concentrations and PLC synergy at higher concentrations. The curves qualitatively appear maybe biphasic. The methods suggest fitting to a biphasic equation, but no data is presented as to the quality of this fit relative to a monophasic fit. Since this is one of the keys to the argument being made this needs to be presented more quantitatively with convincing statistics. Again the argument would be much stronger with direct measurement of PLC reaction products by methods other than BRET.

2) The magnitude of the effect is small once the cAMP-IP3 sensitization component is removed comparing 4 E and F. So it is not clear how physiologically relevant the crosstalk at high ligand concentration is. The authors show crosstalk in native cells (Fig. 2), but how much of the crosstalk in primary cells is due to the Gq-b γ PLC activation crosstalk vs cAMP-IP3 receptor sensitization was not tested.

3) In figure 8 experiments are done with PLC β 3 with disabled auto inhibition. these experiments are confounded by the increase in resting IP3 that may occur when these constitutively active mutants are expressed in cells. This cannot be measured with a BRET reporter. Again, I believe that direct measures of either IP, IP3 or DAG are needed here.

Other points.

4) RFU needs to be better defined. The way this dye is used with a quencher to lower baseline fluorescence is different than many fluorescent Ca²⁺ indicator experiments and should be explained in the methods.

5) The methods give a description of the statistics but I could only find 3 panels where statistics were used, panel 4A, 5C, and 8E and in figure 4A it is not clear what comparison is being made. Many of the figures show robust responses and may not need stats but more quantitative statistical analysis is needed in many places.

6) line 429 talks about a C-terminal PH domain. The PH domain is on the N terminus. The references cited talk about G $\beta\gamma$ binding to the C terminus, which is probably not right. The new cryo EM structures do show G $\beta\gamma$ binding to the N terminal PH domain and the EF hand.

7) Line 563: The authors talk about obligate Gq signaling for PLC β 2 which I do not believe is sufficiently supported by the data. The original data concerning synergistic PLC activation by G α_q and G $\beta\gamma$ reported by Philip et al and others earlier indicate that PLC β 2 is not regulated synergistically. PLC β 2 is highly restricted in its tissue distribution primarily to monocytes

such as neutrophils where it is highly enriched and abundant. It is true that in most cells which contain PLC β 3 it is very hard to see Gi dependent responses without coincident Gq activation, however; In neutrophils, highly enriched in PLC β 2, chemoattractants and chemokines give robust IP3 and Ca $^{2+}$ responses without coincident Gq activation. I do not think the experiment with PLC β 2 expressed in HEK cells is sufficient to say this is a general mechanism for regulation of this isoform since most other data say this is not the way this enzyme is regulated (Fig. 6D).

8) Line 44: in cardiac myocytes Epac-PLC activation does not mobilize cytosolic Ca $^{2+}$, rather it sensitizes RYR2 to Ca $^{2+}$ induced Ca $^{2+}$ release.

We thank all reviewers for constructive feedback, concurring assessment, and helpful suggestions related to our manuscript. In the revised version, we addressed all points raised by the referees. We are convinced that this improved our existing work and manuscript text. All newly added manuscript text is boxed yellow.

REVIEWER COMMENTS

Reviewer #1 (Remarks to the Author):

In this manuscript, Brands et al propose a novel mechanism of Ca²⁺ mobilisation dependent on Gβγ released from the association with Gs. The authors use a calcium mobilisation assay to comprehensively evaluate calcium signalling over a wide number of cellular conditions, including endogenously expressing cells and HEK293 cells with KO of various signalling partners. It is concluded that following Gq priming, Gs-coupled GPCR activation produces activated free Gβγ complexes alter inositol lipid signalling to cause an increase in Ca²⁺ signalling. This new mechanism extends our understanding of the canonical GPCR-calcium signalling and is important for the field.

Response authors: We thank the reviewer for their positive assessment and appreciation of our study. We hope that the below point-to-point reply adequately addresses all questions, concerns, and suggestions.

Major comments:

1. In Figure 1, the authors use PTX to block the native Gi/o activation for panel C and D, but not for β2AR in panel B. As β2AR could also couple to Gi/o, it would be more convincing to collect data for β2AR in presence of PTX.

Response authors: As suggested, we have now collected β2AR Ca²⁺ data in the presence of PTX and provide this new data as **Supplementary Fig. 2**. We did not use PTX at first submission because (i) we obtained no evidence for Gi-Ca²⁺ in the Gs-KO background and (ii) never observed Gi-coupling of endogenous β2AR in HEK293 cells in our earlier studies (e.g. Grundmann et al., NCOMMS, 2018, DOI: 10.1038/s41467-017-02661-3).

The revised manuscript text now includes new data and interpretation: “These Gq-primed, Iso-triggered calcium transients were unaltered by pertussis toxin (PTX)-pretreatment ruling out Gi/o contribution (Supplementary Fig. 2). Gq-primed Iso-Ca²⁺ was undetectable in HEK293 cells lacking Gα_s and Gα_{oif} (hereafter HEK-ΔGs (Stallaert et al., 2017) even after Gq priming, uncovering Gs as essential mediator of the observed Ca²⁺ signals (Fig. 1a_{iii}) and consistent with the absence of Gi/o contribution.”

Supplementary Fig. 2: Iso-triggered $\beta_2\text{AR-Ca}^{2+}$ after Gq priming is not diminished by PTX pretreatment. HEK2993 cells were primed with 100 μM ATP at $t = 20$ s, followed by a second addition at $t = 140$ s of either Iso or Calcium ionophore A23187 in the absence or presence of PTX. Shown are representative traces and concentration effect curves derived from the maximum calcium response of the second addition of Iso on $\beta_2\text{AR}$, as well as bar chart quantification of A23187 (5 μM) after ATP priming. Where indicated, cells were pretreated overnight (16 h) with 100 ng/ml of PTX. Representative traces are mean + SD, averaged data are mean + SEM of three biologically independent experiments, each performed in duplicate.

2. Figure 3, panel C claims that exogenous $\beta_2\text{AR}$ can couple to Gq which requires further support as it contradicts prior observations. The maximum BRET signal change reported in this figure is only $\sim 2\%$ of the baseline, which does not help instilling the confidence in the conclusions either. There are several well-established biosensor assays that could detect G protein coupling. It would be preferred to use unmodified Gq in these assays to avoid artifactual coupling. It would be also nice to see the response of an agonist at a canonical Gq-coupled receptor to compare whether the small BRET ratio reported indicates physiological significance.

Response authors: As suggested, to increase the confidence in Gq-recognition of exogenous $\beta_2\text{AR}$, we have performed a well-established, elegant biosensor assay (Masuho et al., Methods Mol Biol. 2015; doi: 10.1007/978-1-4939-2914-6_8) that utilizes **unmodified Gq**. To further instill confidence in our conclusions, we have performed an **IP1 accumulation** assay and have included an agonist for a canonical Gq-coupled receptor. We find that overexpressed but not endogenous $\beta_2\text{AR}$ promotes detectable IP1 accumulation that is abolished by the Gq-selective inhibitor FR, and which amounts to about 40% of IP1 produced by the bona-fide Gq-coupled M3 receptor. Thus, we use **three complementary and confirmatory experimental approaches** (Fig. 3 c, d^{NEW}, e^{NEW}) to ensure that the conclusions drawn are supported by more than one method. Please also see reference (Inoue et al., Illuminating G-Protein-Coupling Selectivity of GPCRs, Cell 2019, doi: 10.1016/j.cell.2019.04.044) reporting Gq recognition by exogenous $\beta_2\text{AR}$ in Fig.2.

The revised manuscript text now reads: "Direct Gq recognition and activation by exogenous $\beta_2\text{AR}$ is further supported in three distinct ways; with bioluminescence resonance energy transfer (BRET)-based G protein biosensors monitoring activation-induced conformational changes of both modified (Fig. 3c) (Schihada et al., 2021) or unmodified Gq (Fig. 3d) (Masuho et al., 2015), and with IP₁ accumulation assays that serve as a proxy for Gq activation (Fig. 3e). FR completely (Fig. 3c) or partially (Fig. 3d)

ablated the detectable BRET changes and fully reversed the Iso-mediated IP₁ accumulation (Fig. 3e) confirming direct engagement of Gq by exogenous β₂AR in all instances.”

Figure 3. Direct Gq coupling of overexpressed β₂AR eliminates the need for heterologous Gq priming. ... d_i Schematic for the BRET-based Gβγ release assay monitoring freed Gβγ dimers after G protein activation of heterotrimers harboring unmodified Gα subunits. **d_{ii-iii}** Iso-induced BRET increase between Venus-labeled Gβγ and the membrane-associated C-terminal fragment of the G protein-coupled receptor kinase 3 fused to NanoLuciferase (masGRK3ct-NanoLuc), shown as real-time BRET recordings and their bar chart quantification. **e** Inositol monophosphate (IP₁) accumulation measured in naïve HEK293 cells transfected to express the β₂AR. Where indicated, cells were pretreated with FR to silence the function of Gq proteins (1 μM in **a-d**; 10 μM in **e**).

3. In figure 4E, the calcium signal of 1 uM Iso is about 170 RFU. however, the calcium signal for 1 uM Iso is about 200 RFU in figure 1A and 1B and is 80 RFU for w/o in figure 5A. These experiments were done under exact same conditions. What explains these differences which exceed variability shown by error bars?

Response authors: Data for this study have been collected during a period of four years, using HEK293 cells at different times and passages showing variability for both the first Gq peak, which amounts to 350 fluorescence units in Fig. 1Aii, 250 in Fig. 4e, and 450 in Fig. 5a (now Fig. 6a), as well as the second Iso peak. If a first Gq stimulus is particularly efficacious, a second Iso stimulus may be less efficacious, hence explaining why the Iso response is relatively small in Fig. 5a (now 6a). The important point is, however, that the qualitative data incl. their interpretation remains the same, i.e. that Gq priming is required for Iso Ca²⁺ in HEK293 cells at endogenous β₂AR expression levels.

4. Concentration response curves in Figure 4G-H could be fit with a singular sigmoidal curve instead of the “biphasic” curve. More explanation should be given regarding why the “biphasic” curve was used, when it is not apparent from the data. A curve fit with two inflection points is likely to fit data better, as this allows for all data points to be on the curve fit, compared to a larger number of data points that fall outside of the sigmoidal curve fit usually used for this type of data. The “biphasic” curve was found to be integral in determining the mechanism of action, however the reasoning for this curve fitting is not justified. A more elaborate description of what analysis was performed and what constraints were used in the methods section would also help guide the reader.

Response authors: We appreciate and understand the request for a more elaborate description of what analysis was used and what constraints were set. We provide this lacking information in the figure legend (previous Fig. 4g, h is the **new Fig. 5a, b**) and **chose to perform additional experimentation to strengthen the experimental data: quarterstep concentration effect curves (Fig. 5a^{NEW}, b^{NEW})**. We feel that the **new experimental design**, which now includes **many more concentrations** is much more robust and allows to represent much more adequately the two components of a biphasic concentration-effect relationship. We also explain why the biphasic model was used and that no constraints were set to fit the data, in the legend to **Fig. 5a^{NEW}, b^{NEW}**.

The revised manuscript text now reads: “To investigate whether a complementary approach to diminish the overall cAMP-IP₃R impact would also allow to unmask contribution of the cAMP-independent Ca²⁺ release mechanism, we employed Gq priming at low stimulus intensity. Indeed, a two-component concentration-effect relationship emerged exclusively for Iso after priming with both CCh and ATP at single digit micromolar concentrations (Figure 5a, b). We noted that the Iso-mediated high potency Ca²⁺ release response was closely resembled in magnitude by Fsk at a maximally effective concentration (Fig. 5a, b).”...

Figure 5. Fsk is a proxy to discriminate cAMP-dependent from cAMP-independent Ca²⁺ after Gq priming in recombinant and primary cells. a-d Calcium mobilization in HEK293 cells (a, b), primary pre-adipocytes (c), and MEFs (d) following the two consecutive addition protocol. **a, b** Iso- and Fsk-induced cytosolic Ca²⁺ increase in HEK293 cells after priming with solvent, 3 μM ATP (a) or 1 μM CCh (b). ... Data in (a, b) were fit to a biphasic concentration-effect of model to minimize the distance of the measured data points from the predicted data points without using constraints.

5. In figure 4F, iso induced lower maximum calcium signal in HEK-ΔAC3/6 cell; and figure 5A shows that IP₃R inhibitor 2-APB could eliminate Iso-mediated β₂AR-calcium signal after Gq priming. It would be important to apply 2-APB to HEK-ΔAC3/6 cell to see whether it could eliminate Iso-mediated β₂AR-calcium signal, which could strengthen the conclusion that cAMP-independent pathway is IP₃R-dependent.

Response authors: As suggested, we have applied the IP₃R inhibitor 2-APB to HEK- Δ AC3/6 cells (**Supplementary Fig. 12^{NEW}**) and find that pre-treatment of cells with this inhibitor abolishes Iso-mediated β 2AR-calcium signal.

The revised manuscript text now reads: “Because IP₃Rs are a point of convergence for distinct upstream signaling pathways (Gs, Gq, Gi/o- β), we explored their involvement using the IP₃R antagonist 2-APB. Pretreatment of HEK293 cells and of Δ AC3/6 cells with 2-APB eliminated the Iso-mediated β ₂AR-Ca²⁺ after Gq priming but also all Gq-Ca²⁺ evoked by ATP, indicating that IP₃-mediated Ca²⁺ release is an essential step for both stimuli (Fig. 6a, **Supplementary Fig. 12**).”

Supplementary Fig. 12: 2-APB eliminates Iso-mediated β ₂AR-Ca²⁺ after Gq priming in Δ AC3/6 cells. Calcium mobilization in Δ AC3/6 cells following the two consecutive addition protocol. At t = 20 s, the Gq stimulus ATP 100 μ M was added, followed by a second addition at t = 140 s of Iso. Data show representative Iso-induced Ca²⁺ traces and their quantification as concentration-effect-curves in the absence or presence of 50 μ M of the IP₃R antagonist 2-APB after ATP priming. Real-time Ca²⁺ recordings are mean values + SEM of technical duplicates, concentration-effect curves are mean values \pm SEM of four (w/o 2-APB) and three (with 2-APB) independent biological experiments.

6. BRET data showing IP₃ formation in Figure 5C (and also Figure 8E) is also not convincing. From the bar graph in Figure 5C it appears that the difference between buffer (approx. 0.006) and 1 μ M Iso (approx. 0.01) stimulation is approximately 0.004. This is a very small Δ BRET ratio therefore, this experiment would also benefit from having a positive control, to determine whether this small change indicates physiological relevance. Alternatively, a more robust IP₃ formation assay should be used, as the conclusion from this experiment is instrumental to lead to Figures 6-8 in the manuscript.

Response authors: As suggested and because the BRET change is small even with a bona fide Gq-coupled receptor in the original publication (Δ BRET ratio for the overexpressed angiotensin 2 AT1 receptor amounted to 0.015 only in *Gulyas et al., PLOS One, 2015; ref#66* in our manuscript), we assume that the small BRET ratios we obtained at endogenous expression levels do not necessarily indicate lack of physiological relevance *per se* but rather are a sensor-intrinsic determinant. Because extensive optimization attempts to increase the Δ BRET ratios in this experiment including variation of Gq stimuli ATP and CCh at different concentrations did not improve the BRET amplitude for Iso, we **chose to pursue the alternative suggestion of this reviewer** and performed the more robust IP₁ accumulation assay. In this assay we applied Iso and two additional Gs-GPCR stimuli and observed no detectable IP₁ formation for any Gs-GPCR

stimulus but significant enhancement of IP₁ formation when ATP or CCh were coapplied (Fig. 6d^{NEW}). Moreover, because the conclusion from this experiment is instrumental to the following figures, we have developed (Methods), validated (Supplementary Fig. 20^{NEW}) and applied (Fig. 6e^{NEW}) a NanoBit-based biosensor detecting depletion of phosphatidylinositol,4,5-bisphosphate (PIP₂), the immediate consequence of PLCβ hydrolysis upstream of IP₃, DAG, and Ca²⁺. We find Iso-mediated PIP₂ depletion after Gq priming in an FR-sensitive manner. Thus, our revised Fig. 6 (formerly Fig. 5) now contains three complementary and confirmatory experimental approaches (Fig. 6c, d^{NEW}, e^{NEW}) to strengthen the conclusion from this experiment.

The revised manuscript text pertaining to the new data in Fig. 6 now reads: Because IP₃ production is rapid and transient as it is metabolized to IP₂ and IP₁, we also quantified its degradation product IP₁ after accumulation in cells. We detected robust Iso-induced IP₁ accumulation exclusively after Gq priming (Fig. 6d_i). We obtained equivalent results for the two other Gs-GPCR stimuli, PGE₁ and NECA, respectively, both provoking IP₁ accumulation only after Gq priming (Fig. 6d_{ii,iii}). We also observed Iso-mediated reduction of PIP₂ levels, the immediate consequence of PLCβ hydrolysis, in Gq-primed cells and this effect was completely blunted by FR pretreatment (Fig. 6e). These data point to active participation of Gs-GPCRs in plasma membrane phospholipid hydrolysis by stimulation of PLCβ isozymes, key orchestrators of inositol lipid-dependent signaling responses.

Figure 6. Gs-coupled β_2AR drives IP₃ formation, IP₁ accumulation and PIP₂ depletion after Gq priming. ... d_{i-iii} Agonist-induced IP₁ accumulation in naïve HEK293 cells with and without prior ATP (100 μM) or CCh (100 μM) priming using Iso (d_i), PGE₁ (d_{ii}), and NECA (d_{iii}) to stimulate β_2AR , EP₂/EP₄, and A_{2A}/A_{2B}, respectively. e Iso-induced PIP₂ depletion after Gq priming. Cartoon illustration of the PIP₂ hydrolysis NanoBiT-based biosensor. PIP₂ hydrolysis is reflected by rapid translocation of the Small BiT (SmBiT)-tagged PH domain of PLC δ 1 from plasma membrane

localized Large BiT (LgBiT)-CAAX to the cytosol resulting in decreased luminescence. ... Ca^{2+} measurements were performed in duplicate; DMR, IP_1 accumulation and PIP_2 depletion in triplicate, and IP_3 -BRET time-courses were quadruplicate determinations. Statistical significance was calculated with a two-way ANOVA with Dunnett's (c) and Šídák's (d, e) post-hoc analysis.

In relation to the comment that Fig. 8e (now Fig. 9e) is not convincing, we extend this data set by additional measures of PIP_2 depletion under equivalent conditions (Fig. 9e^{NEW}). Thus, we provide two independent and complementary lines of evidence for Gs- $\text{G}\beta\gamma$ -mediated $\text{PLC}\beta$ activation. The adapted manuscript text now reads: "To eliminate this confounding variable and to unambiguously isolate the direct activation of $\text{PLC}\beta 3$ by Gs-derived $\text{G}\beta\gamma$, we quantified PIP_2 depletion, the immediate consequence of PIP_2 hydrolysis as well as formation of IP_3 , an immediate product of the PIP_2 hydrolysis reaction upstream of IP_3R -controlled and ER-liberated Ca^{2+} . Indeed, $\text{G}\beta\gamma$ -regulated $\text{PLC}\beta 3^{\text{F715A}}$ drives both Iso-mediated PIP_2 depletion and IP_3 formation without Gq priming and these effects were nullified by masGRK3ct (Fig. 9e, f).".

Figure 9. $\text{PLC}\beta 3$ variants with disabled autoinhibition empower Iso-mediated Gs- $\beta\gamma$ - Ca^{2+} without Gq priming. ... e Iso-induced PIP_2 depletion in HEK- $\Delta\text{Gq}/11/12/13$ cells transfected to express the PIP_2 hydrolysis NanoBiT-based biosensor along with $\text{PLC}\beta 3^{\text{F715A}}$, $\beta 2\text{AR}$, and masGRK3ct or empty vector DNA as control. ...

7. The case for the exploration and the investigations performed are quite clear but the writing style is very verbose with an abundance of unnecessary complex arguments, flashy statements and stump words. It is advisable for authors to clean it up- make the Introduction concise and to the point, avoid restating motivation and logic in Results and focus them better on design and data analysis and streamline the Discussion section as well. The readers will appreciate it as the key message of the manuscript is quite simple at the end of the day.

Response authors: we have cleaned up the manuscript as suggested, stream-lined and shortened the Introduction by 200 words, focussed the Discussion, and removed restating motivation and logic in the Results section at several instances. Please note that deletions are not visible in the yellow marked version but we have, for example **deleted** the former lines **106-112** (Intro), **119-128** (Intro), **383-387** (Results), **428-437** (Results). We have also included **all the below suggestions** for rephrasing and simplifying.

We agree that the key message is quite simple, but also unexpected, clearly going beyond the prevailing view that does not consider participation of Gs-liberated G $\beta\gamma$ as active transducer. Therefore, a lot of work went into the design and performance of a plethora of experiments incl. generation of PLC β 1-4 KO cells, that in our opinion were required to provide clear evidence for the claims we make in our study.

Minor comments:

1. All abbreviations used should be defined where first mentioned.

Response authors: DONE.

2. Line 32 and line 524 should be “A long-held tenet in inositol lipid signaling that cleavage...”.

Response authors: if we remove the “is”, sentence will be incomplete, because we are removing the verb.

3. The latter part of the sentence on lines 100-103 should be simplified, as it is difficult to determine the meaning of this sentence.

Response authors: DONE, the confusing **neither... nor** term is simplified to **not ...or...**. The sentence now reads: ” The prevailing theory is that hydrolysis of PIP₂ by PLC β isoforms to acutely increase intracellular Ca²⁺ is stimulated by both active G α_q and Gi-liberated G $\beta\gamma$ dimers, the latter of which activate PLC β 2 and PLC β 3 only, but not by active G α_s , Gs-derived G $\beta\gamma$ or G α_i proteins.”

4. Line 277 could be written in more concise way, e.g. “The mechanism of Gq coordinates ”

Response authors: DONE.

Previous sentence “How active Gq coordinates Gs-GPCR calcium at a mechanistic level is unclear at present.”

New sentence “The mechanism of how active Gq coordinates Gs-GPCR calcium is unclear at present.”

5. Justification for why the specific isoforms of AC (AC3 and AC6) were deleted should be provided.

Response authors: DONE, we now explain more clearly why AC3 was deleted in addition to AC6. The manuscript text now reads on page 10: Signaling junctions composed of IP₃Rs and type 6 AC are responsible for delivering the high cAMP concentrations directly to IP₃Rs (Konieczny et al., 2017; Taylor, 2017; Tovey et al., 2010; Tovey et al., 2008). To lower the impact of these junctions and, additionally, the levels of cAMP in response to Gs-GPCR and AC activation, we used HEK293 cells depleted by CRISPR/Cas9 of endogenous AC3 and AC6 (hereafter Δ AC3/6 cells) (Soto-Velasquez et al., 2018). Both AC isoforms are highly abundant in HEK293 cells and largely responsible for the Fsk-stimulated cAMP formation in this cellular background (Soto-Velasquez et al., 2018).

6. The term ‘biphasic’ should be changed, as there are not two phases occurring at different times, rather two modes of action at different concentrations of Iso.

Response authors: we used the term biphasic as it is quite common in pharmacology to describe concentration effect relationships that are composed of more than one component. Biphasic dose-response curve is also the terminology used in our Prism curve fitting software. We rephrased to “**two-component**” rather than “*biphasic concentration effect relationship*” to better reflect the two modes of action.

7. Why was a two-way ANOVA used to assess changes in maximal effect of Iso in the absence and presence of PKI14-22 (Figure 4A)?

Response authors: the use of statistics is irrelevant to the data set in Fig. 4a because we assess the inhibitor’s ability to dampen Iso responses after Gq priming. We have removed the unnecessary statistic in this figure and adapted the legend.

8. In Figure 6 (quantification panel) the concentration response curves of Ca signal in panel B and C are in series with the vehicle point, however in panel D and E the vehicle condition is not in series. What is the reasoning for this? And also, why in panel D and E the bottom of the curve begins at a Ca²⁺ signal of 5 and not at 0.

Response authors: Thank you for spotting this irregularity. All Ca²⁺ curves have been re-evaluated and are now in series with the vehicle point.

9. Reference to Figure 7C, D on line 472 should in fact reference Figure 8.

Response authors: DONE, thank you, corrected. As we included one additional figure, reference is now made to Fig. 9c, d.

10. Please consider rephrasing sentence on line 554-555 for clarity, e.g. “proves the physiological significance of Gs-Gβγ-Ca²⁺ signaling”.

Response authors: In the course of cleaning up and stream-lining the discussion, this sentence has been deleted.

11. Sentence on line 564-568 is very long and hard to understand “Third, although redundant...PLCβ2 and β3 isozymes only.

Response authors: DONE, we now provide two shorter split sentences and rephrased for enhanced clarity.

Third, although redundant at first glance with Gi-Gβγ-Ca²⁺, Gs-Gβγ-Ca²⁺ is distinct because of enhanced abundance of cAMP within the Ca²⁺ detection window. This cAMP directly increases the sensitivity of ER-localized IP₃Rs to IP₃ and enables detection of two molecularly separable Ca²⁺ release pathways for the Gβγ-sensitive PLCβ3 only.

12. There is only two data points on the bar graph in Figure 8B and no error is shown, however in the figure caption it states that all data is from three independent biological experiments. This figure should be updated to include a third dataset.

Response authors: DONE, the figure is updated to include a third dataset.

13. ATCC number should be included in the methods section, as on line 638 it states that HEK293 wild-type cells were used, however on line 651 it states that HEK293A cells were used. Clarity around which cell line was used is needed.

Response authors: DONE, we have indeed used several distinct HEK lines (HEK293, HEK293A, and HEK293T cells) and have updated the methods section and figure legends accordingly to make cell line usage completely transparent.

14. More explanation describing the method to determine the “mean BRET decrease” for BRET experiments should be provided in the methods section.

Response authors: DONE, the method to determine the mean BRET decrease is described with more details for the Gq-CASE BRET assay and the IP3 BRET assay. For the Gq-CASE BRET assay, the method section has been expanded to include: “Briefly, the raw Δ BRET (%) over the three baseline measurements at each time point t was calculated as $((\text{BRET}_t - \text{mean baseline BRET})/\text{mean baseline BRET}) * 100$. The corrected Δ BRET values at each time point were determined by correcting for vehicle-induced changes in BRET, i.e., by subtracting the mean raw Δ BRET values obtained at the same time point in vehicle-treated wells.”

For the IP3 BRET assay, the method section has been adapted to include: “BRET ratios were normalized dividing the mean of the baseline BRET (I_0) by those at each time point (I). The raw Δ BRET (%) at each time point was calculated by subtracting the normalized baseline BRET values and multiplying by 100. The corrected Δ BRET values at each time point were determined by subtracting the mean raw Δ BRET values obtained at the same time point in vehicle-treated wells.”

Reviewer #2 (Remarks to the Author):

This is an important, elegant, and well presented study. The manuscript is a fantastic example of how to carry out a systematic and rigorous study of mechanisms of signal transduction. The focus is on a previously unknown mechanism by which GPCRs mobilize intracellular calcium using G β g released by from GPCR-stimulated Gs heterotrimers. The study reveals the specific mechanistic details how two of the most prevalent second messenger systems (dependent on cAMP or on Ca²⁺) intersect with each other in yet another unexpected manner by using a combination of genetic and pharmacologic manipulations. The conclusions are well supported by the data, which is presented beautifully (I even like choice of color palettes). The work is systematic and comprehensive, and does an excellent job at dissecting different components of the signaling response to Gs-coupled GPCRs, including confirmatory studies in more than the frequently used cell line HEK293. The introduction and discussion sections are very clear, despite the complexity of the topic an extensive background. In summary, I am glad to recommend to publish this work as is without reservations.

Response authors: we are very honored to receive such positive feedback as the manuscript was a tour de force to make an important contribution to the field: the notion that Gs-GPCRs also partake in inositol-lipid signaling using their G $\beta\gamma$ subunits to mobilize intracellular Ca²⁺, a universal second messenger in eukaryotic cells. We are convinced that G $\beta\gamma$ subunits and their signaling should receive much more attention than before and hope that our revised manuscript version will generate uniform enthusiasm to pave the way in that direction.

Reviewer #3 (Remarks to the Author):

This is an **interesting and well controlled** series of experiments examining how Gs coupled G protein coupled receptors lead to increased Ca²⁺ mobilization through obligate crosstalk with G α_q coupled receptors. Evidence is presented showing that G $\beta\gamma$ released from Gs coupled receptors synergizes with G α_q released from G α_q coupled receptors at the level of phospholipase C beta activation to increase PLC activity leading to enhanced IP₃ production and Ca²⁺ release. This follows on previous work from these investigators demonstrating obligate G α_q priming of G $\beta\gamma$ signaling from G α_i coupled receptors, which in turn follows on earlier work demonstrating that activation of PLC β_3 by G α_q and G $\beta\gamma$ subunits is strongly synergistic providing a mechanistic basis for G α_q priming.

The authors characterize a dual mechanism for receptor crosstalk and Ca²⁺ mobilization: at low Gs-GPCR agonist concentration cAMP sensitizes the IP₃ receptor to IP₃; at higher agonist concentration G $\beta\gamma$ -G α_q PLC activation predominates. These results are significant in that they move beyond the prevailing view that G $\beta\gamma$ signaling to PLC and other targets derives solely from G α_i coupled receptors and suggests that G $\beta\gamma$ released from G protein heterotrimers other than G α_i can participate in downstream signaling processes. The overall experiments are very thorough but there are some issues that should be addressed.

Response authors: we are grateful for the appreciation of our study and the constructive criticism to improve our work further. Below please find our responses to each individual comment/concern.

1) The major concern that I have is with the dissection of the two modes of Ca²⁺ release sensitization by Gs coupled receptors that is proposed. As indicated by the authors regulation of Ca²⁺ release is regulated by many factors other than simply phospholipase C activation and Ca²⁺ release is an indirect measure. Thus, the analysis must be conducted very carefully. While the experiments measuring Ca²⁺ are somewhat convincing, to really show synergy at the PLC level, measurements of the PLC reaction products, either IP, IP₃ or DAG would be much more definitive and should applied to a greater extent and with greater rigor than is shown here. To the authors credit they tried to do this using a BRET based IP₃ reporter. While suggestive, I find these figures 5 and 8E less than convincing. There is a small but significant increase with ATP and the addition of Iso marginally increases this. More convincing would be to prime with a concentration

of ATP that gives a very small IP₃ response on its own, which should in turn lead to a more pronounced increase in IP₃ when Iso is added. This should work because 3 μM ATP is able to prime the iso response. That being said an independent method that does not rely on BRET directly measuring IP accumulation, IP₃ production or DAG should be used to more definitively make the case.

To this point, the recognition that cAMP sensitization of IP₃ receptors is one mechanism for Gq-Gs crosstalk is carefully considered here but is indirect. The authors claim that the dose response curves in figures 4G and H, and 6 D and E are biphasic, with cAMP-IP₃ receptor sensitization operating at low ligand concentrations and PLC synergy at higher concentrations. The curves qualitatively appear maybe biphasic. The methods suggest fitting to a biphasic equation, but no data is presented as to the quality of this fit relative to a monophasic fit. Since this is one of the keys to the argument being made this needs to be presented more quantitatively with convincing statistics. Again the argument would be much stronger with direct measurement of PLC reaction products by methods other than BRET.

Response authors: we highly appreciate these numerous valuable comments and have taken the following measures to strengthen our data and hence the conclusions based on them.

First, as suggested, **to really show synergy at the PLC level**, we have measured both PLC reaction products **IP₃** and **IP₁** (**Fig. 6d^{NEW}**) and, additionally, also the first step of the PLCβ hydrolysis reaction, **depletion of phosphatidylinositol,4,5-bisphosphate (PIP₂)**. To this end, we have **developed** (Methods), **validated** (**Supplementary Fig. 20^{NEW}**) and **applied** (**Fig. 6e^{NEW}**, **Fig. 9e^{NEW}**) a **NanoBit-based biosensor detecting PIP₂ depletion**, the immediate consequence of PLCβ hydrolysis upstream of IP₃, DAG, and Ca²⁺.

In the IP₁ accumulation assay we stimulated the cells with Iso and two additional Gs-GPCR ligands and observed no detectable IP₁ formation for any Gs-GPCR stimulus but significant enhancement of IP₁ formation when ATP or CCh were coapplied (**Fig. 6d^{NEW}**). When following PIP₂ depletion over time, we find Iso-mediated PIP₂ depletion after Gq priming in an FR-sensitive manner. Thus, our revised Fig. 6 (formerly Fig. 5) now contains **three complementary and confirmatory experimental approaches** (**Fig. 6c, d^{NEW}, e^{NEW}**) to strengthen the conclusion from this experiment, and provide a more solid mechanistic basis for detection of PLC synergy in the Ca²⁺ assays. Please keep in mind that Ca²⁺ release pathways downstream of Gs-GCRs is a major focus of our study.

The revised manuscript text pertaining to the new data in Fig. 6 now reads: **Because IP₃ production is rapid and transient as it is metabolized to IP₂ and IP₁, we also quantified its degradation product IP₁ after accumulation in cells. We detected robust Iso-induced IP₁ accumulation exclusively after Gq priming (Fig. 6d_i). We obtained equivalent results for the two other Gs-GPCR stimuli, PGE₁ and NECA, respectively, both provoking IP₁ accumulation only after Gq priming (Fig. 6d_{ii, iii}). We also observed Iso-mediated reduction of PIP₂ levels, the immediate consequence of PLCβ hydrolysis, in Gq-primed cells and this effect was completely blunted by FR pretreatment (Fig. 6e). These data point to active**

participation of Gs-GPCRs in plasma membrane phospholipid hydrolysis by stimulation of PLC β isozymes, key orchestrators of inositol lipid-dependent signaling responses.

Figure 6. Gs-coupled β_2AR drives IP_3 formation, IP_1 accumulation and PIP_2 depletion after Gq priming. ... d_{i-iii} Agonist-induced IP_1 accumulation in naïve HEK2993 cells with and without prior ATP (100 μM) or CCh (100 μM) priming using Iso (d_i), PGE $_1$ (d_{ii}), and NECA (d_{iii}) to stimulate β_2AR , EP_2/EP_4 , and A_{2A}/A_{2B} , respectively. e Iso-induced PIP_2 depletion after Gq priming. Cartoon illustration of the PIP_2 hydrolysis NanoBIT-based biosensor. PIP_2 hydrolysis is reflected by rapid translocation of the Small BiT (SmBiT)-tagged PH domain of PLC δ 1 from plasma membrane localized Large BiT (LgBiT)-CAAX to the cytosol resulting in decreased luminescence. ... Ca^{2+} measurements were performed in duplicate; DMR, IP_1 accumulation and PIP_2 depletion in triplicate, and IP_3 -BRET time-courses were quadruplicate determinations. Statistical significance was calculated with a two-way ANOVA with Dunnett's (c) and Šídák's (d, e) post-hoc analysis.

Second, in relation to the very good suggestion to prime with a lower ATP concentration. We have carried out extensive optimization to improve the Δ BRET ratios with the IP_3 BRET-based biosensor. However, neither lowering the ATP concentration nor switching to CCh enabled improved BRET amplitudes for Iso. Lowering the first Gq stimulus allowed detection of a second Gq stimulus but did not enhance the Iso-mediated Δ BRET ratio above 0.003/0.004. Although we are not able to improve the Iso-mediated BRET ratio with this sensor any further, this data together with the more **robust IP_1 accumulation** data (Fig. 6d^{NEW}) and **PIP_2 depletion** data (Fig. 6e^{NEW}) allow “to more definitely make our case”.

Third, in relation to detection of PLC synergy in the Ca^{2+} assays, this reviewer appreciates biphasic curve fitting yet misses information on fit quality. We now include the requested and necessary information in the figure legend (previous Fig. 4g, h is the **new Fig. 5a, b**) and **chose to perform additional experimentation to strengthen the experimental data: quarterstep concentration effect curves (Fig. 5a^{NEW}, b^{NEW})**. We feel that the new experimental design, which now includes many more concentrations is more robust and

allows to represent much more adequately the two components of a biphasic concentration-effect relationship. We also explain why the biphasic model was used and that no constraints were set to fit the data, in the legend to **Fig. 5a^{NEW}, b^{NEW}**. With these new data and the two-component fits, we provide a solid molecular basis for dissection of the two Ca^{2+} release modes.

Figure 5. Fsk is a proxy to discriminate cAMP-dependent from cAMP-independent Ca^{2+} after Gq priming in recombinant and primary cells. a-d Calcium mobilization in HEK293 cells (a, b), primary pre-adipocytes (c), and MEFs (d) following the two consecutive addition protocol. a, b Iso- and Fsk-induced cytosolic Ca^{2+} increase in HEK293 cells after priming with solvent, 3 μM ATP (a) or 1 μM CCh (b). ... Data in (a, b) were fit to a biphasic concentration-effect model to minimize the distance of the measured data points from the predicted data points without using constraints.

We also attempted to strengthen the data in Fig. 6 d, e (now Fig. 7d, e) with additional intermediate concentrations but decided to stop these attempts (new data not requested) as the signal windows don't really permit quantitative pharmacology with additional intermediate concentrations including dissection of parallel signaling mechanisms. Therefore, we chose to fit the new PLC β 2 data set in Fig. 6d (now Fig. 7d) with a monocomponent equation, in agreement with the minimal Ca^{2+} induced by fsk in this series of experiments; the conclusion of this section does not change, i.e. that the qualitative and quantifiable difference between PLC β 1 and β 4 versus PLC β 2 and β 3 isozymes parallels with their natural regulation by G protein β subunits.

2) The magnitude of the effect is small once the cAMP-IP3 sensitization component is removed comparing 4 E and F. So it is not clear how physiologically relevant the crosstalk at high ligand concentration is. The authors show crosstalk in native cells (Fig. 2), but how much of the crosstalk in primary cells is due to the Gq-bg PLC activation crosstalk vs cAMP-IP3 receptor sensitization was not tested.

Response authors: A very good **proxy** of the crosstalk can be obtained by using forskolin (Fsk) as an indicator. The value of Fsk as pharmacological tool to determine the extent of crosstalk stems from careful analysis of the biphasic Ca^{2+} concentration-effect-curves (Fig. 5a^{NEW}, b^{NEW}). Please note the **clear correlation** between the maximum Ca^{2+} amplitude achieved with Fsk and the cAMP-dependent component of the Iso concentration effect curve. We took advantage of this correlation and used Fsk as a proxy to probe the contribution of cAMP to Ca^{2+} responses after Gq priming in our primary cell models (preadipocytes, preACs, and mouse embryonic fibroblasts, MEFs; Fig. 5c^{NEW}, d^{NEW}). We strengthen our Ca^{2+} data providing parallel, real time cAMP quantifications for both Iso and Fsk within the Ca^{2+} detection window in both primary cell systems without and after Gq priming (Fig. 5e_{ii}^{NEW}, e_{iii}^{NEW}). Interestingly, although both stimuli, Fsk and Iso, produce comparable cAMP within the Ca^{2+} detection window, Fig. 5e), cAMP-dependent Ca^{2+} appears to play no role (preACs, Fig. 5c^{NEW}) or only a minor role (MEFs, Fig. 5d^{NEW}) suggesting that PLC β synergy may be exclusively (preACs) or partly responsible (MEFs) for the observed Iso- Ca^{2+} responses.

An entire section and Figure 5^{NEW} is now dedicated to the dissection of Ca^{2+} release mechanisms in primary cells. The new manuscript section is as follows:

Fsk serves as a proxy to discriminate cAMP-dependent from cAMP-independent Ca^{2+} after Gq priming

To investigate whether a complementary approach to diminish the overall cAMP-IP₃R impact would also allow to unmask contribution of the cAMP-independent Ca^{2+} release mechanism, we employed Gq priming at low stimulus intensity. Indeed, a two-component concentration-effect relationship emerged exclusively for Iso after priming with both CCh and ATP at single digit micromolar concentrations (Figure 5a, b). We noted that the Iso-mediated high potency Ca^{2+} release response was closely resembled in magnitude by Fsk at a maximally effective concentration (Fig. 5a, b). Therefore, we used Fsk as a proxy to probe the contribution of cAMP to the Ca^{2+} release mechanisms engaged by Gs-GPCRs in our primary cell models. Interestingly, unlike Iso- Ca^{2+} , which readily emerged after Gq priming in both preACs and MEFs, Fsk- Ca^{2+} was undetectable in the preACs, but detectable in MEFs, yet smaller in amplitude as compared with Iso- Ca^{2+} (Fig. 5c, d and Supplementary Fig. 11). Because robust cAMP formation was observable for both stimuli within the Ca^{2+} detection window under primed and non-primed conditions in both preACs and MEFs (Fig. 5e), and because Fsk-cAMP even surpassed that of Iso in amplitude in preACs (Fig. 5e_{ii}), we interpreted that the absence of detectable Fsk Ca^{2+} in preACs indicates no major contribution of the cAMP-dependent mechanism in this cellular background. Conversely, both cAMP-dependent and cAMP-independent Ca^{2+} release mechanisms are operative in MEFs. From these data, we concluded that (i) the qualitative and quantitative contribution of Gs-GPCR- Ca^{2+} release pathways is cellular-context-dependent, and (ii) Iso- β_2 AR-Gs- Ca^{2+} in HEK293 cells (Fig. 5a, b) is composed of two separable molecular mechanisms, one reliant on cAMP and involving sensitization of IP₃Rs, the other cAMP-independent but otherwise undefined.

Figure 5. Fsk is a proxy to discriminate cAMP-dependent from cAMP-independent Ca²⁺ after Gq priming in recombinant and primary cells. **a-d** Calcium mobilization in HEK293 cells (**a**, **b**), primary pre-adipocytes (**c**), and MEFs (**d**) following the two consecutive addition protocol. **a**, **b** Iso- and Fsk-induced cytosolic Ca²⁺ increase in HEK293 cells after priming with solvent, 3 μM ATP (**a**) or 1 μM CCh (**b**). **c**, **d** Iso- and Fsk-Ca²⁺ in pre-adipocytes (**c**) and MEFs (**d**) after priming with 10 μM 5-HT (**c**) or 1 μM ATP (**d**). Data show representative real-time Ca²⁺ recordings and their quantification as either concentration-effect curves (**a**, **b**) or bar charts (**c**, **d**) including the calcium ionophore A23187 (5 μM). The two rightmost panels in **a** and **b** depict the maximum Ca²⁺ amplitudes of the Iso-mediated high potency Ca²⁺ release along with Fsk at a maximally active concentration. **e** Live cell real-time cAMP imaging in preACs (**e**_{ii}) and MEFs (**e**_{iii}) using the intramolecular FRET-based pcDNA3.1-miCNBD sensor⁶⁵. **e**_i Cartoon illustration of the sensor principle: The sensor contains the cyclic nucleotide binding domain from the bacterial *MlotiK1* channel (miCNBD) flanked by citrine and cerulean at its N- and C-terminus, respectively. At low cAMP abundance both fluorophores are in close proximity (high FRET state) but move further apart upon cAMP increases (low FRET state). FRET changes in response to Iso and Fsk under non-primed and primed conditions in preACs (**e**_{ii}) and MEFs (**e**_{iii}) are means + SEM of the indicated *n* cells. FRET ratios are inverted to show enhanced cAMP abundance as increased FRET ratios. Pooled data are mean values ± SEM of *n* independent biological experiments (**a**: *n* = 5-6; **b**: *n* = 4-

6; **c, d**: $n = 3$), each performed in duplicate. Representative calcium traces are means + SEM. Data in **(a, b)** were fit to a biphasic concentration-effect model to minimize the distance of the measured data points from the predicted data points without using constraints.

We also make reference to these new data in the discussion section (page 23): “Precisely this feature, lack of cAMP-dependent Ca^{2+} after Gq priming in primary preACs led us to speculate that the observed Gs- Ca^{2+} is $\text{G}\beta\gamma$ -dependent also in this primary cell context.”

We also attempted to use additional tools to dissect Iso- Ca^{2+} after Gq priming in primary cells using (i) the small molecule $\text{G}\beta\gamma$ inhibitor gallein (Bonacci et al., Science 2006 (PMID: 16627746), Lehmann et al., MolPharm 2008 (PMID: 18006643)), and (ii) a recently published macrocyclic Gs inhibitor (cpGD20) reported to sequester Gas-GDP in the OFF-state but to prolong $\text{G}\beta\gamma$ activation after receptor stimulation (Dai et al., Cell 2022 (PMID: 36170854)). We hoped to visualize both inhibition and enhancement of $\text{G}\beta\gamma$ signaling but instead observed no inhibition of $\text{G}\beta\gamma$ signals by gallein, and inhibition rather than enhancement of Gs Ca^{2+} after Gq priming by cpGD20. As these validation data are in disagreement with elegant, published literature and pose more questions than answers, we respectfully suggest to not consider such data in the revised manuscript. Certainly, all these “*negative/inconclusive*” data are available to the reviewers from the corresponding author upon request. We also included two additional coauthors, Luna Schmacke (PhD candidate) and Torsten Steinmetzer (Group leader, University of Marburg, Germany) who have synthesized the macrocyclic Gs inhibitor cpGD20 for our revision. Because inhibition of Gs in a nucleotide-selective manner and of $\text{G}\beta\gamma$ is of great value to the community, we will investigate these pharmacological tools further beyond the timeframe and scope of this revision to better understand their mode of action.

3) In figure 8 experiments are done with PLC β 3 with disabled auto inhibition. these experiments are confounded by the increase in resting IP $_3$ that may occur when these constitutively active mutants are expressed in cells. This cannot be measured with a BRET reporter. Again, I believe that direct measures of either IP, IP $_3$ or DAG are needed here.

Response authors: We absolutely agree with this reviewer that experiments with mutant PLC β 3 variants need to carefully take basal activity into account, and hence elevated IP $_3$ levels in the absence of ligand stimulation. Therefore, the data in Fig. 8e (now Fig. 9f) required **careful titration of both the BRET IP $_3$ biosensor and the mutant PLC β 3 variant**. To conceive IP $_3$ BRET data in Fig. 9f, we chose PLC β 3-F715A, which is less basally active as compared with PLC β 3- Δ XY. We succeeded in measuring Iso-mediated BRET, a response that is entirely $\text{G}\beta\gamma$ -mediated. We respectfully suggest maintaining our carefully generated data set in Fig. 9f but extend it by additional measures of PIP $_2$ depletion under equivalent conditions, that we have used earlier to monitor the action of PLC β more directly (Fig. 6e^{NEW}). The new data (Fig. 9e^{NEW}) are fully consistent with the IP $_3$ BRET data in Fig. 9f demonstrating Iso-mediated PIP $_2$ depletion in a $\text{G}\beta\gamma$ -dependent fashion. The adapted results text now reads: “To eliminate this confounding variable and to unambiguously isolate the direct activation of PLC β 3 by Gs-derived $\text{G}\beta\gamma$, we quantified PIP $_2$ depletion, the immediate consequence of PIP $_2$ hydrolysis as well as formation of IP $_3$,

an immediate product of the PIP₂ hydrolysis reaction upstream of IP₃R-controlled and ER-liberated Ca²⁺. Indeed, Gβγ-regulated PLCβ3^{F715A} drives both Iso-mediated PIP₂ depletion and IP₃ formation without Gq priming and these effects were nullified by masGRK3ct (Fig. 9e, f).” With this newly added data set, we enhance the confidence in our conclusions, i.e. that we are recording Gs-Gβγ-dependent signaling, even more because these data were generated in Gαq/11/12/13 KO cells.

Figure 9. PLCβ3 variants with disabled autoinhibition empower Iso-mediated Gs-βγ-Ca²⁺ without Gq priming. ... e Iso-induced PIP₂ depletion in HEK-ΔGq/11/12/13 cells transfected to express the PIP₂ hydrolysis NanoBiT-based biosensor along with PLCβ^{F715A}, β₂AR, and masGRK3ct or empty vector DNA as control. ...

For clarity and enhanced transparency, we also added the following statement in appreciation of the well-documented constitutive activity of the two PLCβ3 mutants to the Discussion section: “Indeed, the lower efficacy of masGRK3ct to diminish Iso-Ca²⁺ in PLCβ3^{ΔXY} as compared with PLCβ3^{F715A} expressing cells parallels with the stronger constitutive activity of the PLCβ3^{ΔXY} variant (Charpentier et al., 2014; Lyon et al., 2011, 2014; Pfeil et al., 2020), and hence a more prevalent contribution of the cAMP-IP₃R axis (Konieczny et al., 2017; Tovey et al., 2008).”

Other points.

4) RFU needs to be better defined. The way this dye is used with a quencher to lower baseline fluorescence is different than many fluorescent Ca²⁺ indicator experiments and should be explained in the methods.

Response authors: As suggested we have significantly expanded the description of the Ca²⁺ assay protocol in the methods section. We now explicitly mention the inclusion of a quencher (masking dye) to reduce baseline fluorescence and to increase the signal-to-noise ratio. This quencher is a component of a commercially available Calcium assay kit that we followed as per manufacturer’s instructions.

5) The methods give a description of the statistics but I could only find 3 panels where

statistics were used, panel 4A, 5C, and 8E and in figure 4A it is not clear what comparison is being made. Many of the figures show robust responses and may not need stats but more quantitative statistical analysis is needed in many places.

Response authors: We have added statistical testing and comparison in all instances where response occurrence is not obvious to the eye or where it is required. The following panels include new statistical analysis: Fig. 3d(iii), Fig. 3e, Fig. 6d, Fig.6e, Fig. 8a and Fig. 8b. In Fig. 4a statistical analysis has been deleted.

6) line 429 talks about a C-terminal PH domain. The PH domain is on the N terminus. The references cited talk about Gbg binding to the C terminus, which is probably not right. The new cryo EM structures do show Gbg binding to the N terminal PH domain and the EF hand.

Response authors: Thank you very much for pointing out this flaw. As we shortened the results section and avoided - where possible - restating of logic and motivation as suggested by reviewer#1, the section including the flawed statement has been deleted.

7) Line 563: The authors talk about obligate Gq signaling for PLCb2 which I do not believe is sufficiently supported by the data. The original data concerning synergistic PLC activation by Gαq and Gβγ reported by Philip et al and others earlier indicate that PLCb2 is not regulated synergistically. PLCb2 is highly restricted in its tissue distribution primarily to monocytes such as neutrophils where it is highly enriched and abundant. It is true that in most cells which contain PLCb3 it is very hard to see Gi dependent responses without coincident Gq activation, however; In neutrophils, highly enriched in PLCb2, chemoattractants and chemokines give robust IP₃ and Ca²⁺ responses without coincident Gαq activation. I do not think the experiment with PLCb2 expressed in HEK cells is sufficient to say this is a general mechanism for regulation of this isoform since most other data say this is not the way this enzyme is regulated (Fig. 6D).

Response authors: Thank you for pointing out the need of enhanced semantical precision. We have rephrased the corresponding Discussion section accordingly:

Former text: In other words, neither Gi- nor Gs-Gβγ-Ca²⁺ signaling modules act as stand-alone entities but only as part of a Gq-dominated PLCβ-dependent network. In this network, PLCβ2 and β3 function as bottleneck allowing to pass Gβγ signals onward only if active Gαq pulls the licensing trigger.

Revised text: "In this network, PLCβ3 functions as bottleneck allowing to pass Gβγ signals onward only if active Gαq pulls the licensing trigger (Pfeil et al., 2020; Sanchez et al., 2022). For PLCβ2, an isoform that is highly abundant in monocytes and neutrophils, chemoattractants and chemokines are known to provoke robust IP₃ and Ca²⁺ responses through Gi-derived Gβγ (Khan et al., 2013), apparently without coincident Gq activation. We speculate that promiscuous G16 proteins, which are highly abundant in cells from the hematopoietic lineage (Amatruda et al., 1991), and which are well known to stimulate PLCβ1-3 isozymes in a manner comparable to that of Gq (Kozasa et al., 1993), may assume the licensing function and substitute for Gq in myeloid precursors."

We thank this reviewer for bringing up the high expression of PLC β 2 in myeloid cells which together with the abundance of G15 or G16 could explain the efficacious Ca²⁺ responses of chemokine and chemoattractant receptors. To the best of our knowledge, this hypothesis has not yet been investigated and is clearly beyond the scope of the current study. However, we hope that the new data in our manuscript will provoke re-analysis of the molecular mechanisms underlying chemokine and chemoattractant signaling in myeloid cells. We don't see such studies in contradiction to earlier elegant PLC β 2 work but rather as extension to the current modes of PLC β 2 regulation.

8) Line 44: in cardiac myocytes Epac-PLC activation does not mobilize cytosolic Ca²⁺, rather it sensitizes RYR2 to Ca²⁺ induced Ca²⁺ release.

Response authors: Thank you for spotting the imprecise wording. The manuscript text now reads: "Among these are activation of protein kinase A (PKA), a main cAMP effector that phosphorylates L-type calcium channels in cardiomyocytes (Christ et al., 2009; Kamp & Hell, 2000), cAMP-EPAC-dependent activation of PLC ϵ (Schmidt et al., 2001), which enhances cytosolic calcium in cardiac myocytes through Ca²⁺-induced Ca²⁺ release (Oestreich et al., 2007), and cAMP-mediated sensitization of IP₃-gated ion channels, which release Ca²⁺ from the ER (Konieczny et al., 2017; Tovey et al., 2010; Tovey et al., 2008)."

REVIEWERS' COMMENTS

Reviewer #1 (Remarks to the Author):

The authors have addressed all my concerns. This is an exciting and important study!

Reviewer #2 (Remarks to the Author):

I was satisfied with the previous version of this manuscript and after reading the response to the other reviewers' comments I remain convinced about the suitability of this work for publication in the journal. This is high quality work that opens new avenues of thinking in the field of GPCR-G protein signaling.

Reviewer #3 (Remarks to the Author):

This is a revised manuscript. All of the points that I have raised were considered carefully and addressed. In particular the new data measuring IP1 and PIP2 hydrolysis address a key point that was raised. This is very thorough and well done.